# Neurostructural subgroup in 4291 individuals with schizophrenia identified using the subtype and stage inference algorithm

Machine learning can be used to define subtypes of psychiatric conditions based on shared biological foundations of mental disorders. Here we analyzed cross-sectional brain images from 4,222 individuals with schizophrenia and 7038 healthy subjects pooled across 41 international cohorts from the ENIGMA, non-ENIGMA cohorts and public datasets. Using the Subtype and Stage Inference (SuStaIn) algorithm, we identify two distinct neurostructural subgroups by mapping the spatial and temporal 'trajectory' of gray matter change in schizophrenia. Subgroup 1 was characterized by an early cortical-predominant loss with enlarged striatum, whereas subgroup 2 displayed an early subcortical-predominant loss in the hippocampus, striatum and other subcortical regions. We confirmed the reproducibility of the two neurostructural subtypes across various sample sites, including Europe, North America and East Asia. This imaging-based taxonomy holds the potential to identify individuals with shared neurobiological attributes, thereby suggesting the viability of redefining existing disorder constructs based on biological factors.

Schizophrenia is one of the most severely disabling psychiatric disorders with a life-time prevalence of 1%; it affects approximately 26 million people worldwide[1]. The etiology of schizophrenia is still not fully understood. Current knowledge implicates multiple neurobiological mechanisms and pathophysiologic processes[2,3]. Furthermore, people diagnosed with schizophrenia show a substantial heterogeneity in clinical symptoms[4], disease progression[5], treatment response[6], and other biological markers[7,8]. In addition, currently available treatments are not aligned with specific pathophysiological pathways/targets, which limits effectiveness of treatment selection[9]. Establishing a new taxonomy by identifying distinct subtypes based on neurobiological data could help resolve some of these heterogeneity-induced challenges. A key goal is to define biological subtypes, based on objective measures derived from imaging and other biomarkers[10].

Artificial intelligence methods such as machine learning can be applied to brain imaging[11] to categorize individuals based on their profiles of brain metrics, and holds the potential for revealing the underlying neurobiological mechanisms associated with disorder subtypes[12]. Machine learning algorithms are increasingly used to subtype brain disorders[13–16]. Prior studies have primarily focused on grouping individuals into distinct categories without considering disease progression[17,18]. A major obstacle to identifying distinct patterns of neuro-pathophysiological progression (referred to as progression subtypes) stems from the lack of sufficient longitudinal data covering the lifespan of the disorder. Recently, a data-driven machine learning approach known as Subtype and Stage Inference (SuStaIn) was introduced[19]. SuStaIn uses a large number of cross-sectional observations, derived from single time-point MRI scans, to identify clusters (subtypes) of individuals with common trajectory of disease

✉e-mail: jffeng@fudan.edu.cn

progression (i.e., the sequence of MRI abnormalities across different brain regions) in brain disorders[20–23]. It should be noted that SuStaIn estimates the pseudo-longitudinal sequence (i.e., SuStaIn trajectory) based on only cross-sectional data. Therefore, the fitted SuStaIn trajectories do not directly reflect the actual pathophysiological progression of the illness. By applying SuStaIn to MRI data from individuals with schizophrenia, primarily collected from the Chinese population, we found that the progression of gray matter loss in schizophrenia can be better characterized through two distinct phenotypes: one characterized by a cortical-predominant progression, originating in the Broca's area/fronto-insular cortex, and another marked by a subcortical-predominant progression, starting in the hippocampus[22]. Such brain-based taxonomies may reflect neurostructural subtypes with shared pathophysiological foundations, with relevance for neurobiological classification[22]. However, the generalizability of the two neurostructural subtypes to diverse populations outside of China, and external validation of the subgrouping is required before applying this knowledge to stratify clinical trials.

The Enhancing Neuro Imaging Genetics through Meta-Analysis (ENIGMA, http://enigma.ini.usc.edu) consortium is dedicated to conducting large-scale analyzes by pooling brain imaging data from research teams worldwide, using standardized image processing protocols. Previously, ENIGMA published findings revealing thinner cerebral cortex, smaller surface area, and altered subcortical volumes in schizophrenia compared to controls[24,25]. Here, we included structural MRI data obtained from 4291 individuals diagnosed with schizophrenia and 7078 healthy controls from 41 international cohorts from ENIGMA schizophrenia groups worldwide and other non-ENIGMA datasets (Supplementary Table 1–2). The large sample size allowed us to conduct systematic and comprehensive analyzes to verify the reproducibility and generality of neurostructural subtypes of schizophrenia across regions/locations and disease stages. This study's aims were: (1) to validate the two neurostructural subtypes with distinct trajectories of neuro-pathophysiological progression in schizophrenia, (2) to verify the reproducibility and generality of the neurostructural subtypes, in subsamples across the world and across disease stages, and (3) to characterize subtype-specific signatures in terms of neuroanatomy and clinical symptomatic trajectory.

Together, these analyzes aim to create an easily accessible (with a single anatomical MRI), interpretable (based on 'progressive' pathology) and robustly generalizable (across ethnic, sex and language differences) taxonomy of subtypes that share common neurobiological mechanisms in schizophrenia. If proven effective, other complex neuropsychiatric disorders with high heterogeneity[26,27], such as major depressive disorder, autism spectrum disorder, and obsessive-compulsive disorder, could also benefit from such a subtyping paradigm. This has the potential to transition the field of psychiatry from syndrome-based to both syndrome- and biology-based stratifications of mental disorders.

## Results

### Two biotypes with distinct pathophysiological progression trajectories

Distinct patterns of spatiotemporal progression of pathophysiological progression were identified using SuStaIn, based on cross-sectional MRI data from 4222 individuals diagnosed with schizophrenia (1683 females, mean age=32.4 ± 11.9 years) and 7038 healthy subjects (3440 females, mean age=33.0 ± 12.6 years) (Table 1). A 2-fold cross-validation procedure resulted in an optimal number of $K = 2$ clusters (subtypes) as determined by the largest Dice coefficient (Fig. 1a), indicating the best consistency of the subtype labeling across all individuals for a model in two independent schizophrenia populations. Figure 1b shows that only 1.2% of people were moved from subtype 1 to subtype 2, and 7.5% were moved from subtype 2 to subtype 1, indicating that 91.3% of individuals' subtype labels were consistent between the SuStaIn

classifications from two non-overlapping data folds. These findings suggest the presence of two stable schizophrenia biotypes with distinct 'trajectories' of pathophysiological progression (here, we put SuStaIn trajectory in quotes as it is not an actual longitudinal trajectory but rather a typical sequence of disease progression reconstructed from cross-sectional data).

Region of interest (ROI)-wise gray matter volume (GMV) z-scores, at each stage of the 'trajectory' for each subtype, show the sequence of regional volume loss across the 17 brain regions for each 'trajectory' (Fig. 1c). To visualize the spatiotemporal pattern of each 'trajectory', z-score whole brain images were mapped to a glass brain template (Fig. 1d). These maps show a progressive pattern of spatial expansion along with later 'temporal' stages of pathological progression distinct for each 'trajectory' (Supplementary Movie 1 and 2). Specifically, 'trajectory' 1 displayed an 'early cortical-predominant loss' biotype. It was characterized by an initial reduction in Broca's area, followed by adjacent fronto-insular regions, then extending to the rest of the neocortex, and finally to the subcortex (Fig. 1d). Conversely, 'trajectory' 2 exhibited an 'early subcortical-predominant loss' biotype where volume loss began in the hippocampus, spread to the amygdala and parahippocampus, and then extended to the accumbens and caudate before affecting the cerebral cortex (Fig. 1d). The two 'trajectories' were highly consistent with our previous findings in a predominantly Chinese schizophrenia cohort[22]. We also re-estimated trajectories based on a validation dataset ($N = 3120$) that has removed the original data used in our previous SuStaIn study[22]. In the validation dataset, we replicated the two 'trajectories' that begin in either the Broca's area or the hippocampus (Supplementary Fig. 1). We also observed a high similarity of 'trajectory' spatiotemporal pattern between the original dataset and the additional dataset ('trajectory' 1, $r = 0.879$, $p < 0.001$; 'trajectory' 2, $r = 0.631$, $p < 0.001$; Spearman correlation test). The phenotypic subtypes, based on the different pathophysiological 'trajectories', are thus replicated in a large cross-geography sample, confirming the presence of two different neuropathological pathways with different anatomical origins in schizophrenia[22].

### Trajectories are repeated in first-episode and medication-naïve samples

The sample size of this study was large enough to allow further exploratory analyses to identify pathophysiological progression trajectories in more homogeneous subsamples of schizophrenia. Here, we re-estimated the SuStaIn 'trajectories' based on a subsample of data from individuals with first-episode schizophrenia with illness duration less than two years ($N = 1122$; 513 females, mean age=25.4 ± 8.6 years), and a subsample of medication-naïve individuals with schizophrenia ($N = 718$, 353 females, mean age = 23.7 ± 7.8 years) (Supplementary Table 3). In both subsamples, we replicated the two 'trajectories' with either the Broca's area or the hippocampus as the sites of origin (Supplementary Fig. 2), indicating that the two initiating regions - ranking ahead of other regional deficits—are the pathological effects of the disease itself, rather than medication-induced effects. Broca's area and the hippocampus may, therefore, be candidate targets for intervention in schizophrenia, as these two brain regions were affected early in the disease process.

### Trajectories are reproducible for samples from different parts of the world

To examine whether the 'trajectories' were reproducible for samples from different parts of the world, we divided all samples into several sub-cohorts based on where the samples were obtained. Here, samples from China, Japan, South Korea and Singapore were classified into the East Asian ancestry (EAS) cohort. Samples from Europe, the United States, Canada and Australia were classified into the European ancestry (EUR) cohorts (Supplementary Table 4). In addition, Chinese, Japanese, European and North American cohorts were further classified by their

**Table 1 | Demographic and clinical characteristics in the primary sample including 4222 schizophrenia patients and 7038 healthy controls**

| | HC($n = 7038$) | | SCZ($n = 4222$) | | SCZ subtype1($n = 2622$) | | SCZ subtype2($n = 1600$) | |
|---|---|---|---|---|---|---|---|---|
| | $n$ | mean(SD) | $n$ | mean(SD) | $n$ | mean(SD) | $n$ | mean(SD) |
| Sex (Female/Male) | 3440/3598 | - | 1683/2539 | - | 1044/1578 | - | 639/961 | - |
| Age (years) | 7038 | 33.0(12.6) | 4222 | 32.4(11.9) | 2622 | 32.4(11.8) | 1600 | 32.4(12.0) |
| Illness duration (years) | - | - | 2333 | 10.5(10.4) | 1442 | 10.4(10.5) | 891 | 10.5(10.4) |
| FES/Chronic/Unknown | - | - | 1112/1623/1477 | - | 696/1002/924 | - | 426/621/553 | - |
| PANSS Positive scale (P1-P7) | - | - | 2651 | 17.2(6.8) | 1622 | 17.3(3.9) | 1029 | 17.0(6.7) |
| PANSS Negative scale (N1-N7) | - | - | 2651 | 17.5(7.6) | 1622 | 17.6(7.6) | 1029 | 17.3(7.6) |
| PANSS General scale (G1-G16) | - | - | 2651 | 34.8(11.6) | 1622 | 35.2(11.6) | 1029 | 34.3(11.6) |
| PANSS Total score | - | - | 2651 | 69.5(22.4) | 1622 | 70.0(22.4) | 1029 | 68.6(22.5) |
| PANSS excitement dimension (P4, P7, G44, G14) | - | - | 1322 | 8.2(3.5) | 823 | 8.2(3.4) | 499 | 8.2(3.5) |
| PANSS depression/anxiety dimension (G1, G2, G3, G6, G15) | - | - | 1322 | 11.3(4.1) | 823 | 11.4(4.1) | 499 | 11.1(4.2) |
| PANSS cognitive dimension (P2, N5, G5, G10, G11) | - | - | 1322 | 10.6(4.0) | 823 | 10.5(4.0) | 499 | 10.6(4.0) |

Abbreviation: HC, healthy control; SCZ, schizophrenia; FES, first-episode schizophrenia; PANSS, Positive and Negative Syndrome Scale.

site locations in terms of geographic distribution (Supplementary Table 4). Such a division was based on the similar ethnic or environmental factors for each country, region, or continent and the size of subsample, which need to be sufficient to conduct a reliable inference of the SuStaIn trajectory. We found that two 'trajectories' (the optimal number was also $K = 2$, which separately re-estimated in each cohort)—with Broca's area leading and the hippocampus leading—were also repeated in EAS (Fig. 2a) and EUR (Fig. 2b) cohorts. In addition, the spatiotemporal pattern of each 'trajectory' showed strong, significant correlations between the EAS and EUR cohorts ('trajectory' 1, $r = 0.948$, $p < 0.001$; 'trajectory' 2, $r = 0.842$, $p < 0.001$; Spearman correlation test). This high level of similarity in the trajectories was also observed between cohorts from other locations (Fig. 2c). This suggests that the two biotypes with distinct 'trajectories' of pathophysiological progression in schizophrenia are robust, and their classification patterns are independent of macro-environmental or ethnogenetic factors.

**Trajectories are associated with neurophysiological, pathological and neuropsychological progressions in schizophrenia**
The SuStaIn calculated the probability of each patient belonging to a specific 'trajectory' and further assigned them to a sub-stage within that 'trajectory'. Individuals who were assigned to the later stages of the 'trajectory' showed a significant correlation with less GMV of Broca's area (Fig. 1e, $r = 0.651$, $p < 0.0001$) and hippocampus (Fig. 1f, $r = 0.615$, $p < 0.0001$). In addition, the later stages were correlated with longer disease duration (Fig. 1g, $r = 0.105$, $p < 0.0001$), worse negative symptoms (Fig. 1h, $r = 0.101$, $p < 0.0001$) and worse cognitive symptoms (Fig. 1i, $r = 0.080$, $p = 0.004$). These results suggest that the SuStaIn 'trajectory' reflects the underlying neural progression in schizophrenia.

**Subtype-specific signatures in neuroanatomical pathology**
To characterize subtype-specific neuroanatomical signatures, we assessed regional morphological measures using FreeSurfer in a subsample including 1840 individuals with schizophrenia and 1780 healthy controls. A total of 330 regional morphological measures in cortical thickness, cortical surface area, cortical volume, subcortical volume and subregion segmentation were quantified (see "Methods").

Regional morphological z-scores (i.e., normative deviations from healthy control group) for each subtype were computed and compared (Fig. 3). Morphological z-scores of all brain regions and inter-subtype comparisons are provided in Supplementary Table 5. Briefly, compared to healthy controls, average cortical volume/area reduction was only observed in subtype 1 (Supplementary Fig. 3a–b), though both subtype 1 and subtype 2 exhibited a moderate reduction in average cortical thickness (Supplementary Fig. 3c). Additionally, largest effects for cortical thickness/volume/area were located within the superior frontal regions for subtype 1 and in the superior/medial temporal regions for subtype2 (Supplementary Table 5). As for subcortical volume, larger effects for volumes of hippocampus, amygdala, thalamus, accumbens and brain stem were observed in subtype 2 compared to subtype 1 (Supplementary Fig. 3d–h). The hippocampal/amygdala subregions with the most significant reduction for subtype 2 were located in the molecular layer and cortico-amygdaloid transition area (Supplementary Fig. 4–5). Interestingly, we observed that, compared to healthy controls, the striatum (i.e., caudate, putamen) was larger among subtype 1 patients and smaller among subtype 2 patients (Supplementary Fig. 3i–j). The difference in the striatum between the two subtypes was also replicated in a subsample of medication-naive individuals with schizophrenia (Supplementary Table 6). The main findings of subtype-specific neuroanatomical signatures are described in Table 2. Taken together, subtype 1 exhibited greater deficits in cortical morphology but enlarged volume of the striatum, whereas subtype 2 displayed more severe volume loss in the subcortical regions, including the hippocampus, amygdala, thalamus, brain stem and striatum.

**Clinical characterization of subtypes**
A total of 2622 (62.1%) individuals with schizophrenia were assigned to subtype 1 and the remaining 1600 patients (37.9%) were assigned to subtype 2. The two subtypes exhibit no significant difference in the age, sex, illness duration or PANSS scores (Table 1). To further characterize the psychotic symptomatic trajectory as the disease progresses for each subtype, we further defined three subgroups according to illness duration (early stage [<2 years], $n = 926$; middle stage [2–10 years], $n = 578$; late stage [>10 years], $n = 682$). The results suggested distinct trajectories of psychotic symptoms between the two subtypes (Fig. 4 and Table 3). Specifically, lower positive symptom severity was observed in late stage patients compared early stage patients in both subtypes (subtype 1, $F = 37.4$, $p = 1.60e − 16$; subtype2, $F = 41.9$, $p = 4.68e − 18$). With the increase of the disease course, subtype 1 showed a gradual worsening of negative symptoms ($F = 4.6$, $p = 9.98e − 3$), whereas the negative symptoms of subtype 2 remained stable across the three stages of the disease course ($F = 0.1$, $p = 0.884$). Additionally, a gradual worsening of depression/anxiety was only observed in subtype 1 ($F = 5.9$, $p = 2.86e − 3$). Inter-subtype

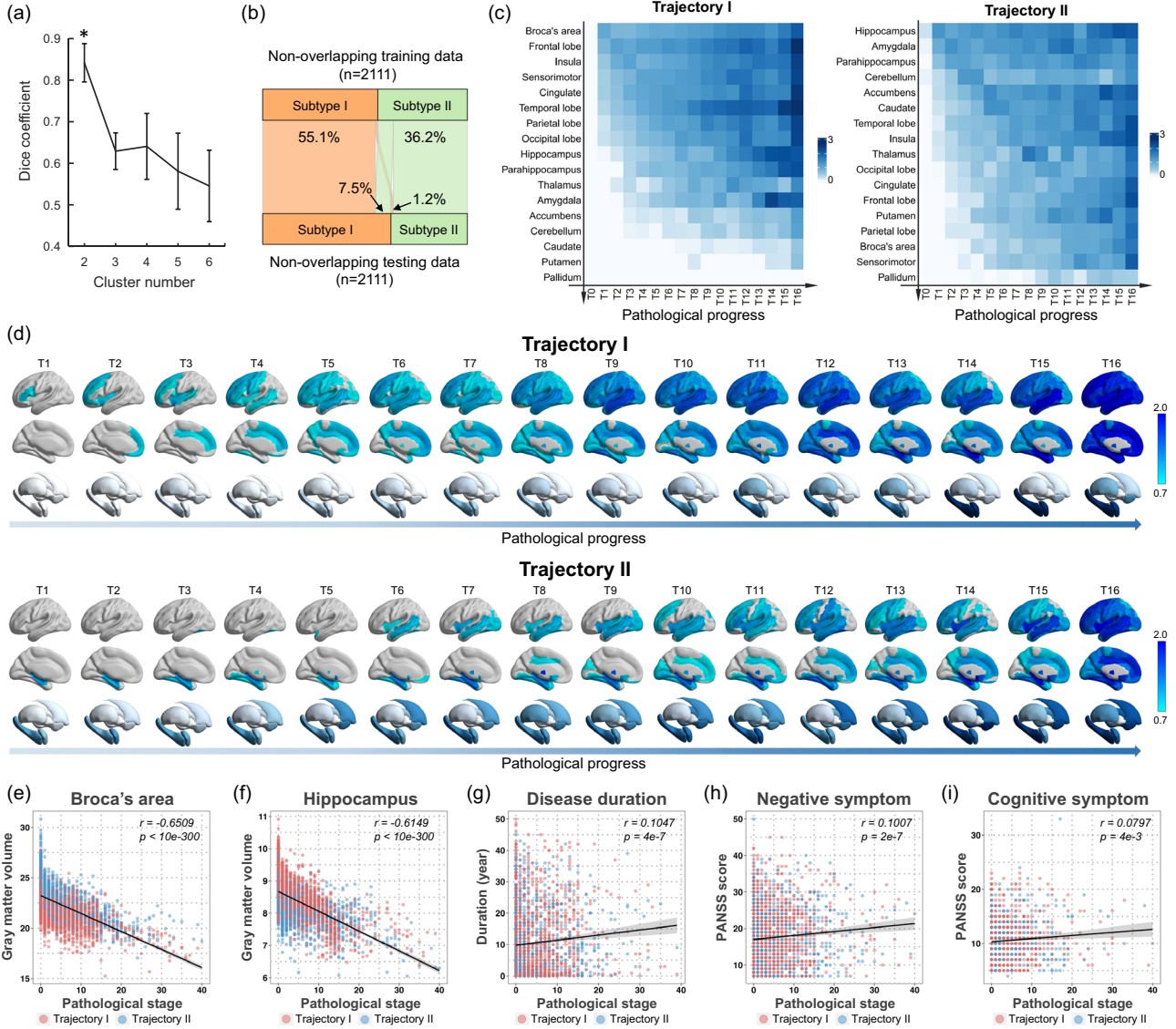

**Fig. 1 | Two pathophysiological progression trajectories in schizophrenia.**
**a** Dice coefficient indicates that $K = 2$ is the optimal number (marked by asterisk) of subtypes with best consistency of the subtype labeling between two independent schizophrenia populations using non-overlap 2-folds cross-validation procedure. This procedure was repeated ten times ($n = 10$) to avoid the occasionality of one split. Data are presented as median values +/- standard deviation (SD). **b** The proportion of individuals whose subtype labels keep consistent by a non-overlap cross-validation procedure. **c** Sequences of regional volume loss across seventeen brain regions for each 'trajectory' via SuStaIn are shown in y-axis. The heatmap shows regional volume loss in which biomarker (y-axis) in a particular 'temporal' stage (T0-T16) in the 'trajectory' (x-axis). The Color bar represents the degree of gray matter volume (GMV) loss in schizophrenia relative to healthy controls (i.e., z score). **d** Spatiotemporal pattern of pathophysiological 'trajectory'. The z-score images are mapped to a glass brain template for visualization. The spatiotemporal pattern of gray matter loss displays a progressive pattern of spatial extension along with later 'temporal' stages of pathological progression that are distinct between trajectories. **e–f** Pathological stages of SuStaIn are correlated with reduced gray matter volume of Broca's area and hippocampus. **g–i** Pathological stages of SuStaIn are correlated with longer disease duration, worse negative symptoms and worse cognitive symptoms. Spearman correlation test is conducted for data analysis in figures (**e–i**). Two-sided p value is reported after multiple comparisons correction by FDR. The error bands in figures (**e–i**) represent 95% confidence interval. $n = 4222$ biologically independent samples in figures (**e–f**). $n = 2333$ biologically independent samples in figure (**g**). $n = 2651$ biologically independent samples in figure (**h**). $n = 1322$ biologically independent samples in figure (**i**).

comparisons showed that at the late stage (illness duration>10 years), subtype 1 exhibited worse positive symptoms ($t = 2.9$, $p = 0.003$), general psychopathology ($t = 2.5$, $p = 0.010$) and worse depression/anxiety ($t = 2.1$, $p = 0.033$) compared to subtype 2, after regressing out the effects of age, sex and SuStaIn stage.

## Generalization of SuStaIn subtyping and staging to unseen cohorts

We investigated whether the SuStaIn subtyping and staging can be generalized to unseen cohorts. A flowchart is shown in Supplementary Fig. 6a. Specifically, the Asian and Europe SuStaIn models were separately built based on the Asian ancestry cohorts and Europe ancestry cohorts, as described in 2.3. The two models were used for subtyping and staging those unseen samples. We compared whether those subtype and stage assignments match the result of the original model that has been built on all cohorts. We observed that most of the unseen individuals can keep the same subtype label with the original model (88.83% for the Asian model; 89.98% for the European model) (Supplementary Fig. 6b). In addition, there was a high consistency of individual staging between stages of unseen data and original model result (Asian model, $r = 0.976$, $p < 0.001$; Europe model, $r = 0.979$, $p < 0.001$, Spearman correlation test) (Supplementary Fig. 6c). These

## (a) East Asian ancestry (EAS) population

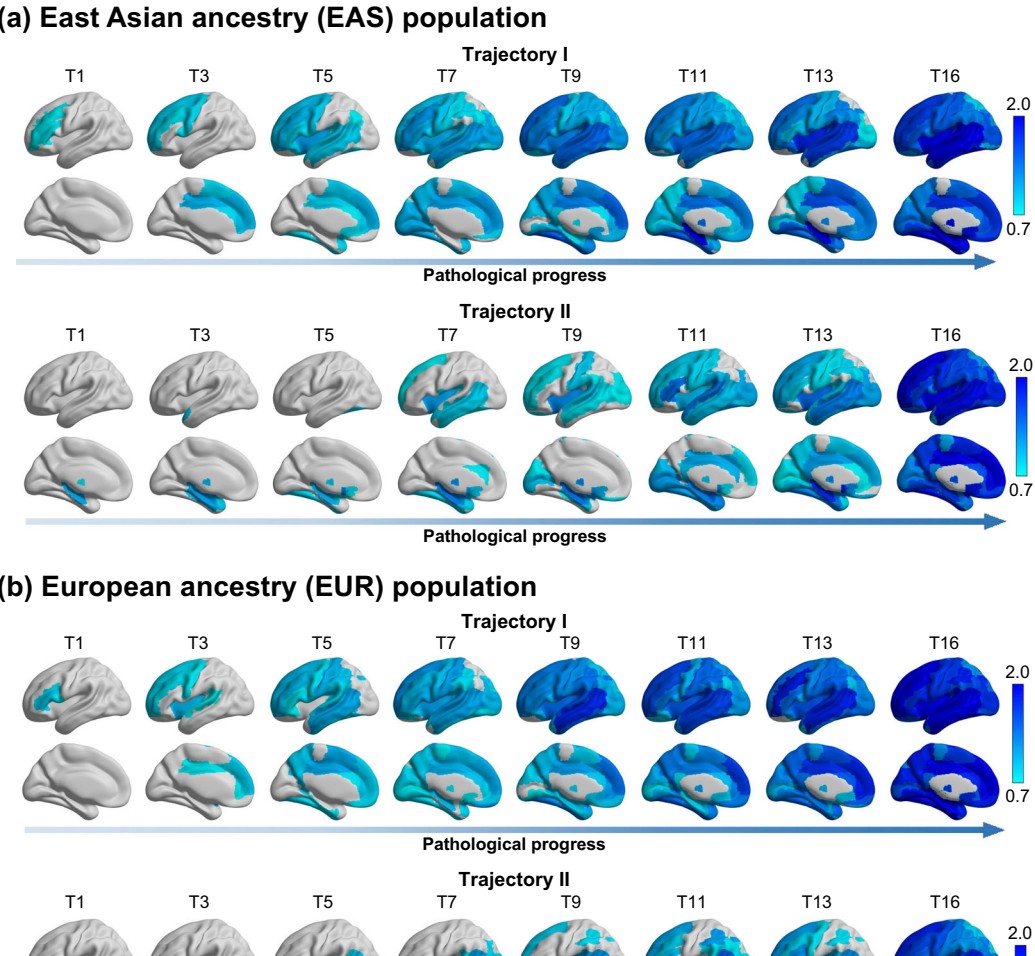

## (b) European ancestry (EUR) population

## (c) Similarity of the trajectories among people from different parts of the world

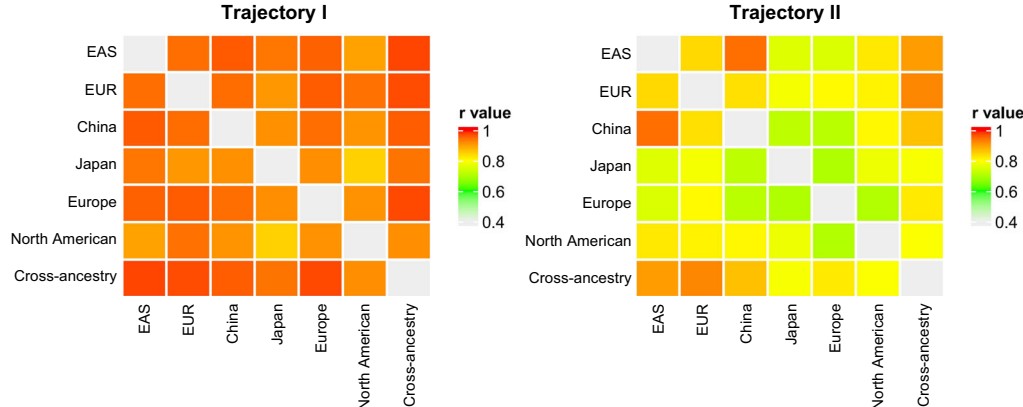

**Fig. 2 | Trajectories are reproducibility for samples from different locations of the world.** Two sets of 'trajectories' are separately derived from two non-overlapping location cohorts, that are (**a**) East Asian ancestry (EAS) cohort, and (**b**) European ancestry (EUR) cohort. The Color bar represents the degree of gray matter volume (GMV) loss in schizophrenia relative to healthy controls (i.e., z-score). **c** The similarity of the spatiotemporal pattern of each 'trajectory' between any two of cohorts is shown by the heatmap. The color bar of the heatmap represents the similarity, which is quantified via the Spearman correlation coefficient between the trajectories from two cohorts. A total of six location cohorts are classified by where the sample locate at, including the EAS, EUR, China, Japan, Europe and North American. The whole sample is labeled as a cross-ancestry cohort.

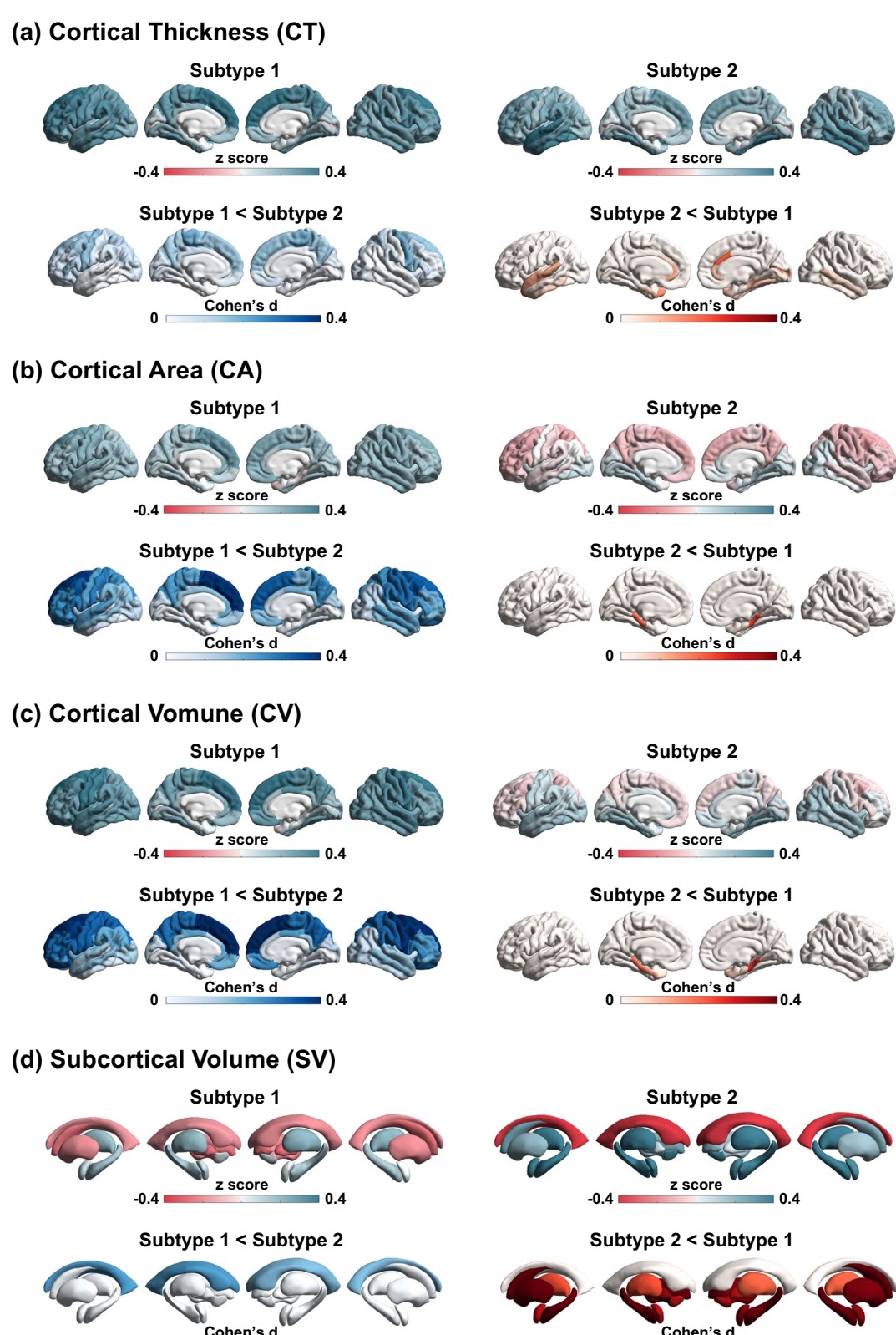

**Fig. 3 | Subtype-specific signatures in neuroanatomical pathology.** Brain morphological measures include (**a**) cortical thickness, (**b**) cortical surface area, (**c**) cortical volume, and (**d**) subcortical volume. For each morphological measure, regional z-scores (i.e., normative deviations from healthy control group) in each subtype are mapped to a brain template for visualization. Effect size of inter-subtype difference is quantified using Cohen's d.

**Table 2 | Main findings of subtype-specific neuroanatomical signatures**

| Morphometry measures | Subtype-specific neuroanatomical signatures |
|---|---|
| Cortical Thickness/Volume/Area | a) Both subtype1 and subtype2 exhibit a moderate degree in the average cortical thickness reduction. |
| | b) Reduction of average cortical volume/area is only observed in the subtype1. |
| | c) The worst reduction of cortical thickness/volume/area is located within the superior frontal regions for the subtype1, but in the superior/medial temporal regions for the subtype2. |
| Subcortical Volume | a) Enlargement of lateral ventricle is found in both subtype1 and subtype2, but much larger in the subtype2. |
| | b) Worse loss volumes of the hippocampus, amygdala, thalamus, and accumbent are observed in the subtype2, compared to the subtype1. |
| | c) Volumes of striatum (i.e., caudate, putamen) are increased in the subtype1, but decreased in the subtype2, compared to the healthy population. |
| Hippocampus segmentation | a) Volume loss in hippocampal subregions is worse in the subtype2, compared to the subtype1. |
| | b) The most significant volume loss is in the molecular layer for the subtype2. |
| Amygdala segmentation | a) The subtype2 shows worse volume loss in amygdala subregions, compared to the subtype1. |
| | b) The most significant decrease in volume is in the cortico-amygdaloid transition area for both the subtypes. |
| Thalamus segmentation | a) The subtype2 shows worse volume loss in thalamus subregions, compared to the subtype1. |
| Brain stem segmentation | a) Volume loss of brain stem subregions is only observed in the subtype2. |

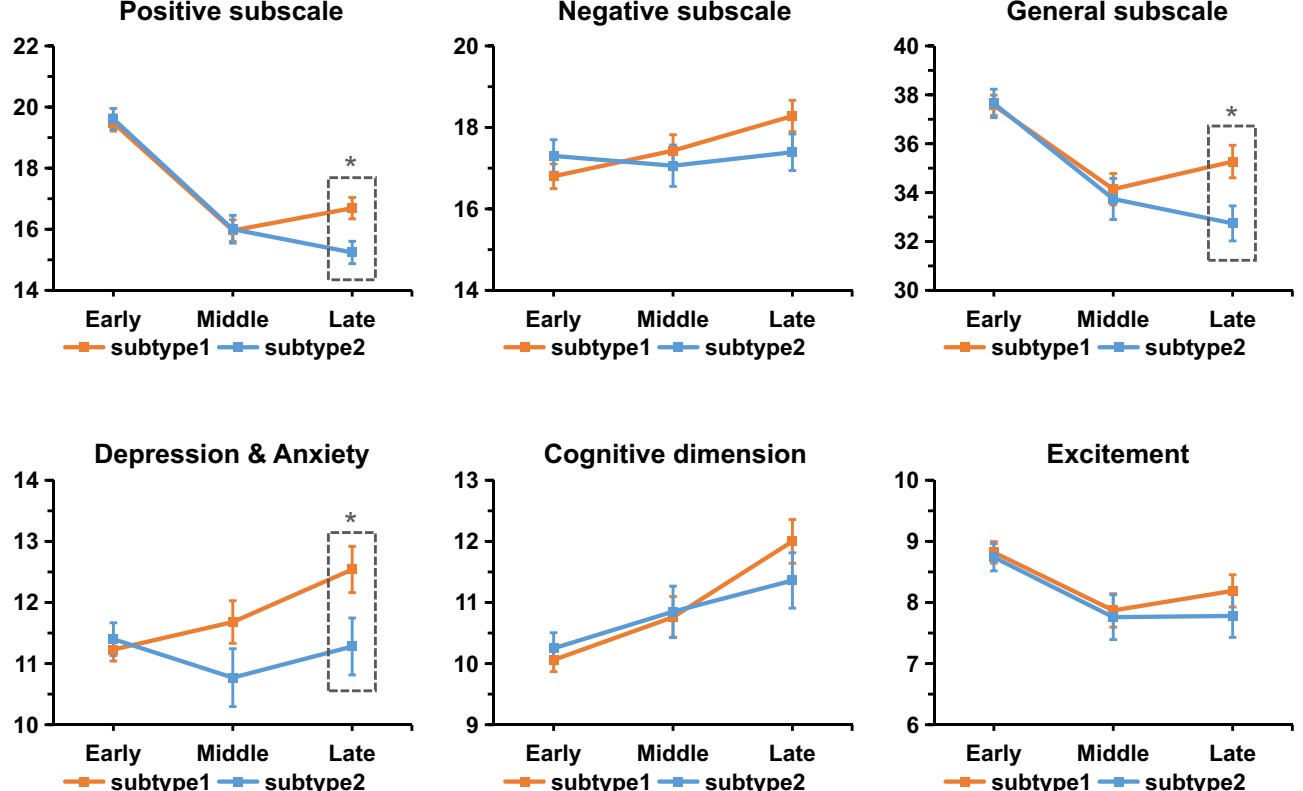

**Fig. 4 | Symptomatic trajectories across three stages of disease duration.** Individuals of each subtype are divided into three subgroups according to their illness durations (early stage: ≤2 years; middle stage: 2–10 years; late stage: >10 years). Two sample $t$ test was performed to compare the inter-subtype difference separately within each of the stages after regressing out the effects of age, sex and SuStaIn stage. * two-sided $p < 0.05$, uncorrected. At the late stage, subtype 1 exhibited worse positive symptom ($t = 2.9$, $p = 0.003$), general psychopathology ($t = 2.5$, $p = 0.010$) and worse depression/anxiety ($t = 2.1$, $p = 0.033$) compared to subtype 2. Data are presented as mean values +/- standard error (se). $n = 579$ (347), 362 (216), and 400 (282) biologically independent samples in the early stage, middle stage and late stage in subtype 1 (subtype 2) for positive, negative and general subscales. $n = 377$ (220), 144 (86), and 166 (109) biologically independent samples in the early stage, middle stage and late stage in subtype 1 (subtype 2) for depression & anxiety, cognitive dimension and excitement dimension.

results indicates a high generalized ability of SuStaIn model to unseen data.

## Discussion

Our study, applying a machine learning algorithm to brain MRI data from over 4000 individuals with schizophrenia, has revealed two distinct neurostructural subtypes based on patterns of neuro-

pathological progression. These subtypes are reproducible and generalizable across different subsamples and illness stages, independent of macroeconomic and ethnic factors that differed across collection locations. Specific patterns of neuroanatomical pathology for each subtype were uncovered. Subtype 1 is characterized by early cortical-predominant loss that first occurs in the Broca's area/fronto-insular cortex, and shows adverse signatures in cortical morphology and an

**Table 3 | Symptom scores for each subtype at different stages of disease duration**

| Symptoms | Subtype 1 | | | F test | | Subtype 2 | | | F test | |
|---|---|---|---|---|---|---|---|---|---|---|
| | Early | Middle | Late | F | p | Early | Middle | Late | F | p |
| PANSS Positive scale (P1–P7) | 19.5(6.4) | 16.0(6.7) | 16.7(7.0)* | 37.4 | 1.60E–16 | 19.6(6.4) | 16.0(6.7) | 15.2(6.2)* | 41.9 | 4.68E–18 |
| PANSS Negative scale (N1–N7) | 16.8(7.3) | 17.4(7.4) | 18.3(7.7) | 4.6 | 9.98E–03 | 17.3(7.4) | 17.1(7.5) | 17.4(7.5) | 0.1 | 0.884 |
| PANSS General scale (G1–G16) | 37.6(10.0) | 34.1(12.2) | 35.3(13.3)* | 10.6 | 2.80E–05 | 37.7(10.8) | 33.7(12.4) | 32.7(12.0)* | 15.6 | 2.30E–07 |
| PANSS Total score | 73.9(19.7) | 67.5(23.2) | 70.2(25.0)* | 9.3 | 9.40E–05 | 74.5(20.9) | 66.7(23.4) | 65.4(22.4)* | 15.7 | 2.05E–07 |
| PANSS excitement dimension (P4, P7, G44, G14) | 8.8(3.4) | 7.9(3.3) | 8.2(3.4) | 4.9 | 8.01E–03 | 8.7(3.3) | 7.8(3.4) | 7.8(3.7) | 4.12 | 0.017 |
| PANSS depression/anxiety dimension (G1, G2, G3, G6, G15) | 11.2(3.7) | 11.7(4.2) | 12.5(4.9)* | 5.9 | 2.86E–03 | 11.4(4.0) | 10.8(4.4) | 11.3(4.9)* | 0.7 | 0.511 |
| PANSS cognitive dimension (P2, N5, G5, G10, G11) | 10.1(3.7) | 10.8(4.0) | 12.0(4.6) | 13.5 | 1.74E–06 | 10.3(3.8) | 10.9(3.9) | 11.4(4.7) | 2.8 | 0.061 |

*indicates significant difference between the subtype1 and subtype2 using two sample t test (two-sided p < 0.05, uncorrected), after regressing out the effects of age, sex and SuStaIn stage. n = 579 (347), 362 (216), and 400 (282) biologically independent samples in the early, middle and late stage in subtype 1 (subtype 2) for PANSS positive, negative and general subscales and total score. n = 377 (220), 144 (86), and 166 (109) biologically independent samples in the early, middle and late stage in subtype 1 (subtype 2) for PANSS depression & anxiety, cognitive dimension and excitement dimension.

enlarged striatum. In contrast, subtype 2 is marked by early subcortical-predominant loss that first appears in the hippocampus, and displays significant volume loss in subcortical regions, including the hippocampus, amygdala, thalamus, brain stem and striatum. Additionally, we observed distinct trajectories of specific symptoms clusters in these two subtypes: as disease progresses, subtype 1 exhibited a gradual worsening of negative and depression/anxiety symptoms, and less of a decline in positive symptoms compared to subtype 2.

Despite the growing body of evidence pointing to group-level gray matter volume deficits in various brain regions - especially in frontal and temporal regions - as well as altered subcortical volume in schizophrenia[28], substantial individual variations persist within this population[8,29]. These inter-individual differences in brain structure may stem from two primary sources of variation. First, differences in underlying etiology and pathogenesis could result in varying clinical characteristics (referred to as phenotypic heterogeneity)[3,30]. Second, relative differences among subjects in the stage of dynamic progression (known as temporal heterogeneity) could further increase differences in the clinical presentation[31,32]. Such variations suggest that the pathological progression of schizophrenia might not be attributed to a single unified pathophysiological process. Indeed, our neurostructural subtypes uncovered two SuStaIn trajectories of gray matter loss through brain structural imaging. Several studies also reported dynamic patterns of accelerated gray matter loss over time in individuals with schizophrenia[33,34]. In addition, the staging of SuStaIn trajectory within the subtype reflects the underlying neurophysiological, pathological, and neuropsychological progressions in schizophrenia. Furthermore, we demonstrated that the phenotypic difference in the intrinsic neuro-pathophysiological trajectory was reproducible across samples worldwide, independent of macroeconomic and ethnic factors that differed across these sites.

The Broca's area/fronto-insular cortex and hippocampus are identified separately in subtype 1 and subtype 2 as the first regions to show gray matter deficits. This is consistent with our prior finding based on individuals with schizophrenia primarily collected from the Chinese population[22]. Furthermore, the current study replicates the same two primary regions in a medication-naïve and a first-episode cohort, suggesting that these neuropathological changes are a reflection of the disease process, rather than medication effects. Broca's area and the fronto-insular cortex have been extensively implicated in schizophrenia[35], supporting Crow's linguistic primacy hypothesis[36] and a triple-network model of the disorder[37]. Abnormalities in Broca's area and related regions have been linked with hallucinations in schizophrenia[38,39]. The early involvement of Broca's area in the pathology could be related to the presence of these core symptoms of

schizophrenia. Moreover, in individuals with psychosis, reductions in the inferior frontal cortex preceding the initial psychotic episode have been reported[40,41]. A prior study reported reduced dopamine release in the prefrontal cortex in patients with schizophrenia[42]. In relation to hippocampal pathology, research has emphasized the hippocampus as one of the initial regions to display volumetric loss in schizophrenia[25,43]. The hippocampus is thought to be involved in potential glutamatergic dysfunction in schizophrenia[3]. Decreased levels of the NMDA co-agonist D-serine were linked to neurobiological alterations similar to those seen in schizophrenia, including hippocampal volume loss[44]. Gray matter loss in schizophrenia is associated with medication, stress, drug use and inactivity[45,46]. In addition, schizophrenia is related to dopaminergic dysregulation, disturbed glutamatergic neurotransmission and increased proinflammatory status of the brain[45]. The causal interrelationships between these processes and gray matter loss are still unclear. These findings offer evidence regarding the specific neuroanatomical locations where gray matter loss is observed in the schizophrenia subtypes. These two potential origins could also offer a viewpoint on the pathological 'spread' of the disorder.

The subtyping method exhibits high potential for distinguishing neurostructural subtypes with shared pathophysiological foundations. Notably, subtype 1 displayed larger volume of the striatum, while subtype 2 demonstrated reduced volume. This was consistent with a previous study, which also identified two anatomical subtypes of schizophrenia: one shows enlarged volume in the basal ganglia; whereas the other shows widespread volumetric reduction in the cortical and some subcortical areas relative to healthy controls[15]. The striatum plays a key role in the dopamine system, which contributes to psychotic symptoms[47]. Nevertheless, studies of striatal pathology have reported inconsistent differences between patients and controls[3]. The variability of the striatum is greater in patients than in controls, which relates to overall structural morphometry[28], dopamine D2 receptor and transporter levels[48]. This indicates that differences might exist within subgroups of the disorder[3]. Alterations in striatal activation are associated with reward-related deficits in schizophrenia[49]. A previous study suggests that disrupted putamen-cortices connectivity during reward-related processing is directly linked to structural changes in the putamen[50]. Despite the unclear causal relationship, this suggests that the differential effects on striatal volume between the two subtypes may be related to striatal dysfunction in schizophrenia. In addition, it is still uncertain whether the discrepancy in striatum between cases and controls indicates a primary pathology or an effect of antipsychotic treatment[3]. Interestingly, this study's subtype-specific striatal differences were replicated in a subset of individuals who had not received antipsychotic treatment, suggesting that striatal

variability persists even in those without antipsychotic treatment. In addition, a recent study reveals a more pronounced and widespread pattern of thinner cortex in deficit schizophrenia, a clinically defined subtype with primary, enduring negative symptoms, compared to non-deficit schizophrenia[51]. A recent work also reveals that the neuro-structural signature with cortical reduction was associated with pro-gressive illness course, worse cognitive performance and elevated schizophrenia polygenic risk scores[52]. This also suggests the existence of distinct subtypes distinguished by unique neuroimaging features. Taken together, our neurostructural subtyping differentiated sub-groups with unique pathological features, thereby enhancing our understanding of the neurobiological mechanisms underlying schizophrenia.

The two identified subtypes may have several potential ther-apeutic implications. While the underlying mechanisms associated with a subtype-specific symptomatic trajectory remain unclear, our research shows divergent long-term clinical outcomes between the two neurostructural subtypes. As the disease advanced, for subtype 1, the negative and depression/anxiety symptoms gradually worsened; for subtype 2 these symptoms remained stable. In addition, subtype 1 experienced worse positive symptoms than subtype 2 at the late stage of disease (i.e., duration > 10 years). This is consistent with a prior study that reported greater gray matter reduction in frontal regions in treatment-resistant compared with treatment-responsive individuals with schizophrenia[53]. Another intriguing aspect is that our prior research on treatment-resistant schizophrenia demonstrated that electroconvulsive therapy (ECT) can substantially enhance the volume of the hippocampus and insula; this is also associated with psychotic symptom alleviation[54–56]. Notably, these two brain regions were also identified as the 'origins' of gray matter loss separately in each subtype. This observation raises the possibility of exploring neuromodulation interventions, such as transcranial magnetic stimulation (TMS), to target these specific brain regions.

This study has several limitations. First, while the SuStaIn algo-rithm estimates pathophysiological trajectories from cross-sectional MRI data, it remains crucial to validate these outcomes with long-itudinal data to verify the brain changes with disease progression over time. Second, the current study benefits from a large sample size, but the inclusion of data from various sites could potentially be influenced by confounding factors, including diverse cohorts, scanners, and locations. Harmonization methods have been employed to alleviate disparities across MRI acquisition protocols. Nonetheless, it remains essential to collect a sufficiently large sample from multi-centers under a standard imaging protocol and experimental paradigm. The lack of cognitive evaluation limits to examine the association of neuro-structural biotype with cognitive impairment in schizophrenia. Third, a substantial portion of individuals with schizophrenia were likely to have received or currently use medications, and data from medication-naïve/free individuals were only available for a subset of the datasets. One important limitation is the assumption of progressive pathology in schizophrenia (discrete events of tissue loss or continuous down-ward drift), when applying SuStaIn. The few existing very long-term imaging studies in schizophrenia support this stance[57] but selection bias cannot be fully overcome in the recruitment process for neuroi-maging studies. Routine anatomical MRI for every person with psy-chosis seeking help, with periodic repeats, may provide better view of the validity of progressive pathology in the future. The selection of z-score waypoints and maximum z-score used in the SuStaIn algorithm should be careful based on prior information about degree of progress in different diseases. The computational complexity of the SuStaIn algorithm is highly time-consuming, which limits the exploration of spatiotemporal patterns of trajectories at finer spatial resolutions.

In summary, our study reveals two distinct neurostructural schi-zophrenia subtypes based on patterns of pathological progression of gray matter loss. We extend the reproducibility and generalizability of these brain imaging-based subtypes across illness stages, medication treatments and different sample locations worldwide, independent of macroeconomic and ethnic factors that differed across these sites. The identified subtypes exhibit distinct signatures of neuroanatomical pathology and psychotic symptomatic trajectories, highlighting the heterogeneity of the neurobiological changes associated with disease progress. This imaging-based taxonomy shows potential for the identification of homogeneous subsamples of individuals with shared neurobiological characteristics. This may be a first crucial step in the transition from only syndrome-based to both syndrome- and biology-based identification of mental disorder subtypes in the near future.

## Methods
### Study samples
This study analyzed cross-sectional T1-weighted structural MRI data from a total of 4,291 individuals diagnosed with schizophrenia (1,709 females, mean age=32.5 ± 11.9 years) and 7,078 healthy controls (3,461 females, mean age=33.0 ± 12.7 years). These datasets came from 21 cohorts of ENIGMA schizophrenia working groups from various countries around the world, 11 cohorts collected from Chinese hospi-tals over the last -10 years, and 9 cohorts from publicly available datasets, i.e., HCP-EP[58], JP-SRPBS[59], fBIRN[60], MCIC[61], NMorphCH[62], NUSDAST[63], DS000030[64], DS000115[65] and DS004302[66]. The datasets came from various countries around the world. Details of demo-graphics, geographic location, clinical characteristics, and inclusion/exclusion criteria for each cohort may be found in the Supplementary Information (Supplementary Table 1–2).

The severity of symptoms was evaluated by the Positive and Negative Syndrome Scale (PANSS)[67], including a positive scale (total score of P1-P7), a negative scale (total score of N1-N7), a general psy-chopathology scale (total score of G1-G16) and total score. In addition, phenotypic characteristics were further quantified in three dimen-sions, such as cognitive (total score of P2, N5, G5, G10, G11), depres-sion/anxiety (total score of G1, G2, G3, G6, G15) and excitement (total score of P4, P7, G44, G14) via a five-factor model of schizophrenia[68].

All sites obtained approval from their local institutional review boards or ethics committees, and written informed consent from all participants and/or their legal guardians. The present study was car-ried out under the approve from the Medical Research Ethics Com-mittees of Fudan University (Number: FE222711).

### Image acquisition, processing and quality control
T1-weighted structural brain MRI scans were acquired at each study site. We used a standardized protocol for image processing using the ENIGMA Computational Anatomy Toolbox (CAT12) across multiple cohorts (https://neuro-jena.github.io/enigma-cat12/). These protocols enable region-based gray matter volume (GMV) measures for image data based on the automated anatomical (AAL3) atlas[69]. Further details of image acquisition parameters and quality control may be found in Supplementary Table 1–2.

### Data harmonization
The ROI-wise GMV measures were first adjusted by regressing out the effects of sex, age, the square of age, site and total intracranial volume (TIV) using a regression model[22]. Subsequently, a harmonization pro-cedure was performed using the ComBat algorithm for correcting multi-site data[70]. The adjusted values were transformed as z-scores (i.e., normative deviations) relative to the healthy control group. We multiplied these z-scores by -1 so that the z-score increases as regional GMV decreases. Finally, we removed these samples if they were marked as a statistical outlier (if any of their regional volumes >5 standard deviations away from the group-level average). After the quality control, 11,260 individuals were included, of which 4222 were schizophrenia patients (1683 females, mean age=32.4 ± 12.4 years) and 7038 healthy subjects (3440 females, mean age=33.0 ± 12.4 years).

### Disease progress modeling

To uncover diverse patterns of pathophysiological progression from cross-sectional only MRI data and cluster individuals into groups (subtypes), we employed a machine learning approach—Subtype and Stage Inference (SuStaIn)[19]. The methodology of SuStaIn has been described in detail previously[19]. We also describe the applicability of SuStaIn algorithm to schizophrenia in Supplementary Materials. Here, we briefly describe the main parameter choices specific to the current study. The SuStaIn model requires an $M \times N$ matrix as input. M represents the number of cases ($M = 4222$). $N$ is the number of biomarkers ($N = 17$). 17 gray matter biomarkers were previously used for SuStaIn modeling in schizophrenia[22]. Here, all of the AAL3 regions of whole brain were separated and merged into 17 regions of interest (ROIs)[22], including frontal lobe, temporal lobe, parietal lobe, occipital lobe, insula, cingulate, sensorimotor, Broca's area, cerebellum, hippocampus, parahippocampus, amygdala, caudate, putamen, pallidum, accumbens and thalamus (Supplementary Table 7). We further examine the relationship between regional volume and illness duration in patients with schizophrenia using the Spearman correlation test (Supplementary Fig. 7). To keep consistent with our previous study[22], we used the z-score thresholds ($z = 1, 2, 3$) as "waypoints" of severity in the SuStaIn model. The maximum z-score in the SuStaIn algorithm was defined at $z = 5$ according to maximum z-score for each biomarker (Supplementary Table 8). We also performed a replication analysis with a reduced maximum z-score ($z = 4$) (Supplementary Fig. 8). We then ran the SuStaIn algorithm with 25 start points and 100,000 Markov Chain Monte Carlo (MCMC) iterations[19] to estimate the most likely sequence that describes spatiotemporal pattern of pathophysiological progression (i.e., 'trajectory').

First, we used the Hopkins statistics to establish whether the data is clustered. A high value ($H = 0.7756$) shows a high clustering tendency at 90% confidence level, supporting a robust existence of clusters. SuStaIn can identify diverse trajectories of pathophysiological progression given a subtype number $K$. We fitted the model for $K = 2$-6 subtypes ('trajectories'), separately. The optimal number of subtypes was determined according to the reproducibility of individual subtyping via a two-fold cross-validation procedure[22]. Specifically, all individuals were randomly split into two non-overlapping folds. This above procedure was repeated ten times. For each fold, we trained the SuStaIn model. For each individual, the trained SuStaIn model provides a subtype label. We measured the consistency of the subtype labeling across all individuals between two folds by using the Dice coefficient. The largest Dice coefficient was obtained for $K = 2$ (see Fig. 1a), indicating the best consistency based on cross-validation. Finally, the two-cluster model of SuStaIn was fitted to the entire sample. The most probable sequence (i.e., the order of biomarkers) was evaluated for each 'trajectory' via SuStaIn. For each individual, SuStaIn calculated the probability (ranging from 0 to 1) of belonging to each 'trajectory', and assigned the individual into a substage of the maximum likelihood 'trajectory' through MCMC iterations. We also estimated the SuStaIn 'trajectories' based on a subsample from individuals with first-episode schizophrenia whose illness duration was less than two years ($N = 1122$, 513 females, mean age=25.4 ± 8.6 years), and a subsample of medication-naïve individuals with schizophrenia ($N = 718$, 353 females, mean age=23.7 ± 7.8 years).

### Visualization of pathophysiological progression trajectory

To visualize the spatiotemporal patterns of pathophysiological progression, we calculated the mean z-score of regional GMV across individuals belonging to the same substage of each SuStaIn 'trajectory'. The images of ROI-wise GMV z-scores were mapped into a glass brain template via visualization tools implemented in ENIGMA Toolbox (https://enigma-toolbox.readthedocs.io/en/latest/index.html) and BrainNetViewer (https://www.nitrc.org/projects/bnv/).

To examine whether the SuStaIn stage (a continuous indicator of the 'temporal' stage of SuStaIn 'trajectory') is associated with pathological processes and clinical characteristics in schizophrenia, we performed Spearman correlations between the SuStaIn stages and the degree of brain atrophy (i.e., regional GMV) in schizophrenia. We also examined whether SuStaIn stages were linked to disease duration, severity of symptoms, and phenotypic characteristics.

### Neuroanatomical signatures using regional morphological measures

To further characterize the neuroanatomical signatures associated with each subtype, we conducted regional morphological analyzes in a subsample including 1840 individuals with schizophrenia and 1780 healthy controls. Brain morphological measures, such as cortical thickness, cortical surface area, cortical volume and subcortical volume, were quantified using FreeSurfer (version 7.3, http://surfer.nmr.mgh.harvard.edu/). A total of 68 × 3 regional measures for cortical thickness, cortical surface area and cortical volume were extracted based on the DK atlas[71], along with 14 subcortical regions (bilaterally nucleus accumbens, amygdala, caudate, hippocampus, pallidum, putamen and thalamus) and 2 lateral ventricles. In addition, we performed an automated subregion segmentation (https://surfer.nmr.mgh.harvard.edu/fswiki/SubregionSegmentation) for the hippocampal substructures ($n = 38$ subregions)[72], the nuclei of the amygdala ($n = 18$)[73], the thalamic nuclei ($n = 50$)[74], and the brain stem structures ($n = 4$)[75], yielding a total of 110 subregional volumetric measures.

Regional morphological measures for each individual with schizophrenia were adjusted by regressing out the effects of sex, age, the square of age, TIV and site, and then transformed to z-scores (i.e., normative deviations from healthy control group). The mean regional morphological z-score across individuals belonging to each subtype was calculated, and mapped to brain templates for visualization of neuroanatomical signature deviation for each subtype relative to healthy population. To further manifest subtype-specific signature in neuroanatomical pathology, we compared the regional morphological z-scores between the two subtypes using two sample t-tests. Multiple comparisons were corrected by family wise error (FWE) correction.

### Distinct symptom profiles between subtypes

To characterize the psychotic symptomatic trajectory with disease duration increases for each subtype, we further divided the individuals of each subtype into three subgroups according to their illness durations (early stage: <2 years; middle stage: 2-10 years; late stage: >10 years). The particular choice of bins was determined according to the distribution of illness duration (early stage $n = 926$, middle stage $n = 578$, late stage $n = 682$) and the size of subgroup enough to perform an inter-subtype comparison. We compared the difference of symptoms among the three stages of disease in each subtype using ANOVA. In addition, two sample t tests were performed to compare the inter-subtype differences separately within each of the stages after regressing out the effects of age, sex and SuStaIn stage.

### Reporting summary

Further information on research design is available in the Nature Portfolio Reporting Summary linked to this article.

## Data availability

The raw image and clinical data are protected and are not available due to data privacy laws. The processed data are available through the following links. Data of NMorphCH, FBIRN and NUSDAST were obtained from the SchizConnect, a publicly available website (http://www.schizconnect.org/documentation#by_project). The NMorphCH dataset and NUSDAST dataset were download through a query interface at the SchizConnect (http://www.schizconnect.org/queries/new). The FBIRN dataset was download from https://www.nitrc.org/projects/

fbirn/. The DS000115 dataset was download from OpenfMRI database (https://www.openfmri.org/). The DS000030 dataset was available at https://legacy.openfmri.org/dataset/ds000030/. The DS004302 dataset was available at https://openneuro.org/datasets/ds004302/versions/1.0.1. The HCP-EP dataset was available at https://www.humanconnectome.org/study/human-connectome-project-for-early-psychosis/. The Japanese SRPBS Multi-disorder MRI Dataset was available at https://bicr-resource.atr.jp/srpbsopen/. Requests for ENIGMA data can be applied via the ENIGMA Schizophrenia Working Group (https://enigma.ini.usc.edu/ongoing/enigma-schizophrenia-working-group/). The statistical data generated in this study are provided in the Supplementary Information/Source Data file. Source data are provided in this paper.

## Code availability

SuStaIn algorithm is available on the UCL-POND GitHub (https://github.com/ucl-pond/). T1-weighted images were processed using the Computational Anatomy Toolbox for Standardized Processing of ENIGMA Data (https://neuro-jena.github.io/enigma-cat12/). A protocol for the current data processing is available at https://docs.google.com/document/d/1lb9v0v4j_OrgAKDh6_9fl3Hz2Wcfg46c/edit/. FreeSurfer (version 7.3, http://surfer.nmr.mgh.harvard.edu/) was used to quantify various morphological measures, such as cortical thickness, cortical surface area, cortical volume and subcortical volume. The visualization of ROI-wise z-score images was conducted using BrainNetViewer (https://www.nitrc.org/projects/bnv/). Other custom codes developed in the current study are available at GitHub (https://github.com/YuchaoJiang91/ENIGMA-SCZ-SuStaIn-Subtype).

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

## Acknowledgements

This work was supported by the grant from Science and Technology Innovation 2030-Brain Science and Brain-Inspired Intelligence Project (No. 2022ZD0212800 to YJ). This work was supported by National Natural Science Foundation of China (No. 82202242 to YJ, 82071997 to WC). This work was supported by the projects from China Postdoctoral Science Foundation (No. BX2021078 and 2021M700852 to YJ). This work was supported by the Shanghai Rising-Star Program (No. 21QA1408700 to WC) and the Shanghai Sailing Program (22YF1402800 to YJ) from Shanghai Science and Technology Committee. This work was supported by National Key R&D Program of China (No. 2019YFA0709502 to JF, 2022ZD0208500 to DY). This work is supported by the CAMS Innovation Fund for Medical Sciences (no. 2019-I2M-5-039 to CLuo). This work was supported by the grant from Shanghai Municipal Science and Technology Major Project (No. 2018SHZDZX01 to JF), ZJ Lab, Shanghai Center for Brain Science and Brain-Inspired Technology, and the grant from the 111 Project (No. B18015 to JF).

## Author contributions

Y.J. and J.F. led the project. P.M.T., J.A.T., and T.G.M.E. provided crucial contributions to the organization and cooperation of the project via ENIGMA. Y.J. and J.F. take responsibility for the integrity of the data and the accuracy of the data analysis in the study. Study concept and design 1: Y.J. and J.F. Acquisition, analysis, interpretation of data 2: Y.J., C.Luo., J.W., L.P., X.C., S.X., J.Z., M.D., H.H., C.G., K.N., K.M., R.H., L.T.W., G.R., S.F.C., N.P., O.A.A., T.K., I.N., Fr.S., F.T.O., L.T., P.U., U.D., T.H., Dd.G., S.M., P.L., Y.T., T.Z., C.Li., W.Y., Y.Z., X.Y., E.Z., C.P.L., S.J.T., A.L.R., Da.G., G.P., J.B., A.K., E.P.C., R.S., P.F.C., M.A.G.L., G.S., F.P., D.V., N.B., J.C., Z.L., J.Y., A.S.G., O.U., B.B.B., A.U.D., K.R.M., V.D.C., K.S., M.G., Y.Q., Y.C.C., W.S.K., S.R.S., C.D., I.S.R., F.I., Ad.B., An.B., M.C., A.B., Si.C., G.P., M.T., M.T.M.P., M.K., F.G., S.K., T.V.R., S.R., M.H., W.W., Se.C., P.S., E.R., Fi.S., A.S., D.T., P.H., S.H., W.O., G.C., D.D.N., A.P., S.T., N.J., L.B.C., D.Y., J.A.T., T.G.M.E., W.C., and J.F. Drafting of the original manuscript 3: Y.J and L.P. Crucial advice to the revision of manuscript and the study 4: L.P., P.M.T., J.A.T., T.G.M.E., W.C., and J.F. All authors contributed edits and approved the contents of the manuscript.

## Competing interests

LP reports personal fees from Janssen Canada, Otsuka Canada, SPMM Course Limited UK and the Canadian Psychiatric Association; book royalties from Oxford University Press; and investigator-initiated educational grants from Sunovion, Janssen Canada and Otsuka Canada, outside the submitted work. TK received unrestricted educational grants from Servier, Janssen, Recordati, Aristo, Otsuka, neuraxpharm. PH has received grants and honoraria from Novartis, Lundbeck, Mepha, Janssen, Boehringer Ingelheim, Neurolite outside of this work. OAA is a consultant to Cortechs.ai and received speakers honorarium from Lundbeck, Janssen, Sunovion. Other authors disclose no conflict of interest.

## Additional information

Yuchao Jiang [1,2], Cheng Luo [3,4,5], Jijun Wang[6], Lena Palaniyappan [7], Xiao Chang [1,2], Shitong Xiang [1,2], Jie Zhang [1,2], Mingjun Duan[3], Huan Huang[3], Christian Gaser [8,9,10], Kiyotaka Nemoto [11], Kenichiro Miura [12], Ryota Hashimoto [12], Lars T. Westlye [13,14,15], Genevieve Richard [13,14,15], Sara Fernandez-Cabello [13,14,15], Nadine Parker [14], Ole A. Andreassen [14], Tilo Kircher[16], Igor Nenadić [16], Frederike Stein[16], Florian Thomas-Odenthal [16], Lea Teutenberg[16], Paula Usemann[16], Udo Dannlowski[17], Tim Hahn [17], Dominik Grotegerd[17], Susanne Meinert [17,18], Rebekka Lencer [17,19,20], Yingying Tang [6], Tianhong Zhang [6], Chunbo Li[6], Weihua Yue [21,22,23], Yuyanan Zhang [21], Xin Yu [21], Enpeng Zhou[21], Ching-Po Lin[24], Shih-Jen Tsai [25], Amanda L. Rodrigue[26], David Glahn[26], Godfrey Pearlson[27], John Blangero[28], Andriana Karuk [29,30], Edith Pomarol-Clotet[29,30], Raymond Salvador[29,30], Paola Fuentes-Claramonte[29,30], María Ángeles Garcia-León[29,30], Gianfranco Spalletta [31], Fabrizio Piras [31], Daniela Vecchio[31], Nerisa Banaj [31], Jingliang Cheng[32], Zhening Liu[33], Jie Yang[33], Ali Saffet Gonul[34], Ozgul Uslu[35], Birce Begum Burhanoglu[35], Aslihan Uyar Demir [34], Kelly Rootes-Murdy[36], Vince D. Calhoun [36], Kang Sim[37,38,39], Melissa Green [40], Yann Quidé [41], Young Chul Chung[42,43,44], Woo-Sung Kim[42,44], Scott R. Sponheim [45,46,47], Caroline Demro[46], Ian S. Ramsay[46], Felice Iasevoli[48], Andrea de Bartolomeis[48], Annarita Barone [48], Mariateresa Ciccarelli [48], Arturo Brunetti[49], Sirio Cocozza [49], Giuseppe Pontillo [49], Mario Tranfa[49], Min Tae M. Park [50,51], Matthias Kirschner [52,53], Foivos Georgiadis[53], Stefan Kaiser[52], Tamsyn E. Van Rheenen[54,55], Susan L. Rossell[55], Matthew Hughes [55], William Woods [55], Sean P. Carruthers[55], Philip Sumner[55], Elysha Ringin[56],

Filip Spaniel[56], Antonin Skoch [56,57], David Tomecek[56,58,59], Philipp Homan [60,61], Stephanie Homan [62,63], Wolfgang Omlor[60], Giacomo Cecere[60], Dana D. Nguyen[64], Adrian Preda[65], Sophia I. Thomopoulos[66], Neda Jahanshad[66], Long-Biao Cui [67], Dezhong Yao [3,4,5], Paul M. Thompson[66], Jessica A. Turner [68], Theo G. M. van Erp [69,70], Wei Cheng [1,2,71,72,73], ENIGMA Schizophrenia Consortium*, Jianfeng Feng [1,2,73,74,75,76,77] ✉ZIB Consortium

[1]Institute of Science and Technology for Brain Inspired Intelligence, Fudan University, Shanghai, China. [2]Key Laboratory of Computational Neuroscience and Brain Inspired Intelligence (Fudan University), Ministry of Education, Shanghai, China. [3]The Clinical Hospital of Chengdu Brain Science Institute, MOE Key Lab for Neuroinformation, School of life Science and technology, University of Electronic Science and Technology of China, Chengdu, China. [4]High-Field Magnetic Resonance Brain Imaging Key Laboratory of Sichuan Province, Center for Information in Medicine, University of Electronic Science and Technology of China, Chengdu, China. [5]Research Unit of NeuroInformation (2019RU035), Chinese Academy of Medical Sciences, Chengdu, China. [6]Shanghai Key Laboratory of Psychotic Disorders, Shanghai Mental Health Center, Shanghai Jiao Tong University School of Medicine, Shanghai, China. [7]Douglas Mental Health University Institute, Department of Psychiatry, McGill University, Montréal, Canada. [8]Department of Psychiatry and Psychotherapy, Jena University Hospital, Jena, Germany. [9]Department of Neurology, Jena University Hospital, Jena, Germany. [10]German Center for Mental Health (DZPG), Site Jena-Magdeburg-Halle, Magdeburg, Germany. [11]Department of Psychiatry, Division of Clinical Medicine, Institute of Medicine, University of Tsukuba, Tsukuba, Japan. [12]Department of Pathology of Mental Diseases, National Institute of Mental Health, National Center of Neurology and Psychiatry, Kodaira, Japan. [13]Department of Psychology, University of Oslo, Oslo, Norway. [14]NORMENT Centre, Division of Mental Health and Addiction, Oslo University Hospital & Institute of Clinical Medicine, University of Oslo, Oslo, Norway. [15]KG Jebsen Centre for Neurodevelopmental Disorders, University of Oslo and Oslo University Hospital, Oslo, Norway. [16]Department of Psychiatry and Psychotherapy, Philipps Universität Marburg, Rudolf-Bultmann-Str. 8, Marburg, Germany. [17]Institute for Translational Psychiatry, University of Münster, Münster, Germany. [18]Institute for Translational Neuroscience, University of Münster, Münster, Germany. [19]Department of Psychiatry and Psychotherapie and Center for Brain, Behavior and Metabolism, Lübeck University, Lübeck, Germany. [20]Institute for Transnational Psychiatry and Otto Creutzfeldt Center for Behavioral and Cognitive Neuroscience, University of Münster, Münster, Germany. [21]Peking University Sixth Hospital, Peking University Institute of Mental Health, NHC Key Laboratory of Mental Health (Peking University), National Clinical Research Center for Mental Disorders (Peking University Sixth Hospital), Beijing, PR China. [22]Chinese Institute for Brain Research, Beijing, PR China. [23]PKU-IDG/McGovern Institute for Brain Research, Peking University, Beijing, PR China. [24]Institute of Neuroscience, National Yang Ming Chiao Tung University, Taipei, Taiwan. [25]Department of Psychiatry, Taipei Veterans General Hospital, Taipei, Taiwan. [26]Department of Psychiatry, Boston Children's Hospital, Harvard Medical School, Boston, MA, USA. [27]Olin Neuropsychiatry Research Center, Institute of Living, Hartford, CT, USA. [28]Department of Human Genetics and South Texas Diabetes and Obesity Institute, School of Medicine, University of Texas of the Rio Grande Valley, Brownsville, TX, USA. [29]FIDMAG Germanes Hospitalàries Research Foundation, Barcelona, Spain. [30]Centro de Investigación Biomédica en Red de Salud Mental, Instituto de Salud Carlos III, Madrid, Spain. [31]Neuropsychiatry Laboratory, Department of Clinical Neuroscience and Neurorehabilitation, IRCCS Santa Lucia Foundation, Rome, Italy. [32]Department of MRI, The First Affiliated Hospital of Zhengzhou University, Zhengzhou, China. [33]National Clinical Research Center for Mental Disorders, Department of Psychiatry, The Second Xiangya Hospital of Central South University, Changsha, Hunan, PR China. [34]Ege University School of Medicine Department of Psychiatry, SoCAT Lab, Izmir, Turkey. [35]Ege University Institute of Health Sciences Department of Neuroscience, Izmir, Turkey. [36]Tri-institutional Center for Translational Research in Neuroimaging and Data Science (TReNDS) [Georgia State University, Georgia Institute of Technology, Emory University], Atlanta, GA, USA. [37]West Region, Institute of Mental Health, Singapore, Singapore. [38]Yong Loo Lin School of Medicine, National University of Singapore, Singapore, Singapore. [39]Lee Kong Chian School of Medicine, Nanyang Technological University, Singapore, Singapore. [40]School of Clinical Medicine, University of New South Wales, SYD, Australia. [41]School of Psychology, University of New South Wales, SYD, Australia. [42]Department of Psychiatry, Jeonbuk National University Hospital, Jeonju, Korea. [43]Department of Psychiatry, Jeonbuk National University, Medical School, Jeonju, Korea. [44]Research Institute of Clinical Medicine of Jeonbuk National University-Biomedical Research Institute of Jeonbuk National University Hospital, Jeonju, Korea. [45]Minneapolis VA Medical Center, University of Minnesota, Minneapolis, MN, USA. [46]Department of Psychiatry and Behavioral Sciences, University of Minnesota, Minneapolis, MN, USA. [47]Department of Psychology, University of Minnesota, Minneapolis, MN, USA. [48]Section of Psychiatry - Department of Neuroscience, University "Federico II", Naples, Italy. [49]Department of Advanced Biomedical Sciences, University "Federico II", Naples, Italy. [50]Department of Psychiatry, Temerty Faculty of Medicine, University of Toronto, TO, Canada. [51]Centre for Addiction and Mental Health, TO, Canada. [52]Division of Adult Psychiatry, Department of Psychiatry, University Hospitals of Geneva, Geneva, Switzerland. [53]Department of Psychiatry, Psychotherapy and Psychosomatics, Psychiatric Hospital University of Zurich, Zurich, Switzerland. [54]Melbourne Neuropsychiatry Centre, Department of Psychiatry, University of Melbourne, MEL, Australia. [55]Centre for Mental Health and Brain Sciences, School of Health Sciences, Swinburne University, MEL, Australia. [56]National Institute of Mental Health, Klecany, Czech Republic. [57]MR Unit, Department of Diagnostic and Interventional Radiology, Institute for Clinical and Experimental Medicine, Prague, Czech Republic. [58]Institute of Computer Science, Czech Academy of Sciences, Prague, Czech Republic. [59]Faculty of Electrical Engineering, Czech Technical University in Prague, Prague, Czech Republic. [60]Psychiatric Hospital, University of Zurich, Zurich, Switzerland. [61]Neuroscience Center Zurich, University of Zurich & Swiss Federal Institute of Technology Zurich, Zurich, Switzerland. [62]Department of Psychiatry, Psychotherapy and Psychosomatics, Psychiatric University Hospital Zurich, Zurich, Switzerland. [63]Experimental Psychopathology and Psychotherapy, Department of Psychology, University of Zurich, Zurich, Switzerland. [64]Department of Pediatrics, University of California Irvine, Irvine, CA, USA. [65]Department of Psychiatry and Human Behavior, University of California Irvine, Irvine, CA, USA. [66]Imaging Genetics Center, Stevens Neuroimaging and Informatics Institute, Keck School of Medicine, University of Southern California, Los Angeles, CA, USA. [67]Department of Clinical Psychology, Fourth Military Medical University, Xi'an, PR China. [68]Psychiatry and Behavioral Health, Ohio State Wexner Medical Center, Columbus, OH, USA. [69]Clinical Translational Neuroscience Laboratory, Department of Psychiatry and Human Behavior, University of California Irvine, Irvine Hall, room 109, Irvine, CA, USA. [70]Center for the Neurobiology of Learning and Memory, University of California Irvine, 309 Qureshey Research Lab, Irvine, CA, USA. [71]Shanghai Medical College and Zhongshan Hospital Immunotherapy Technology Transfer Center, Shanghai, China. [72]Department of Neurology, Huashan Hospital, Fudan University, Shanghai, China. [73]Fudan ISTBI—ZJNU Algorithm Centre for Brain-Inspired Intelligence, Zhejiang Normal University, Jinhua, China. [74]MOE Frontiers Center for Brain Science, Fudan University, Shanghai, China. [75]Zhangjiang Fudan International Innovation Center, Shanghai, China. [76]School of Data Science, Fudan University, Shanghai, China. [77]Department of Computer Science, University of Warwick, Coventry, UK. *Lists of authors and their affiliations appear at the end of the paper. ✉e-mail: jffeng@fudan.edu.cn

## ENIGMA Schizophrenia Consortium

Yuchao Jiang [1,2], Christian Gaser [8,9,10], Kiyotaka Nemoto [11], Kenichiro Miura [12], Ryota Hashimoto [12], Lars T. Westlye [13,14,15], Genevieve Richard [13,14,15], Sara Fernandez-Cabello [13,14,15], Nadine Parker [14], Ole A. Andreassen [14], Tilo Kircher [16], Igor Nenadić [16], Frederike Stein [16], Florian Thomas-Odenthal [16], Lea Teutenberg [16], Paula Usemann [16], Udo Dannlowski [17], Tim Hahn [17], Dominik Grotegerd [17], Susanne Meinert [17,18], Rebekka Lencer [17,19,20], Amanda L. Rodrigue [26], David Glahn [26], Godfrey Pearlson [27], John Blangero [28], Andriana Karuk [29,30], Edith Pomarol-Clotet [29,30], Raymond Salvador [29,30], Paola Fuentes-Claramonte [29,30], María Ángeles Garcia-León [29,30], Gianfranco Spalletta [31], Fabrizio Piras [31], Daniela Vecchio [31], Nerisa Banaj [31], Ali Saffet Gonul [34], Ozgul Uslu [35], Birce Begum Burhanoglu [35], Aslihan Uyar Demir [34], Kelly Rootes-Murdy [36], Vince D. Calhoun [36], Kang Sim [37,38,39], Melissa Green [40], Yann Quidé [41], Young Chul Chung [42,43,44], Woo-Sung Kim [42,44], Scott R. Sponheim [45,46,47], Caroline Demro [46], Ian S. Ramsay [46], Felice Iasevoli [48], Andrea de Bartolomeis [48], Annarita Barone [48], Mariateresa Ciccarelli [48], Arturo Brunetti [49], Sirio Cocozza [49], Giuseppe Pontillo [49], Mario Tranfa [49], Matthias Kirschner [52,53], Foivos Georgiadis [53], Stefan Kaiser [52], Tamsyn E. Van Rheenen [54,55], Susan L. Rossell [55], Matthew Hughes [55], William Woods [55], Sean P. Carruthers [55], Philip Sumner [55], Elysha Ringin [56], Filip Spaniel [56], Antonin Skoch [56,57], David Tomecek [56,58,59], Philipp Homan [60,61], Stephanie Homan [62,63], Wolfgang Omlor [60], Giacomo Cecere [60], Dana D. Nguyen [64], Adrian Preda [65], Sophia I. Thomopoulos [66], Neda Jahanshad [66], Paul M. Thompson [66], Jessica A. Turner [68], Theo G. M. van Erp [69,70], Wei Cheng [1,2,71,72,73] & Jianfeng Feng [1,2,73,74,75,76,77] ✉

## ZIB Consortium

Yuchao Jiang [1,2], Xiao Chang [1,2], Shitong Xiang [1,2], Jie Zhang [1,2], Wei Cheng [1,2,71,72,73] & Jianfeng Feng [1,2,73,74,75,76,77] ✉

