## [Peer Review File · Nature Communications]

Neurostructural subgroup in 4291 individuals with schizophrenia identified using the Subtype and Stage Inference algorithmREVIEWER COMMENTS

Reviewer #1 (Remarks to the Author):

Jiang et al. use data from ENIGMA and other datasets to validate their earlier work (Jiang et al. Nature Mental Health 2023) on identifying two subtypes of schizophrenia with distinct progression patterns using the Subtype and Stage Inference (SuStaIn) algorithm.

The key results are:

1. The algorithm identifies two subtypes, as seen in their previous study. These two subtypes consistently emerge from (i) the whole dataset, (ii) first-episode and medication-naive subsets, (iii) individuals of East Asian ancestry and European ancestry.
2. The stages assigned by the algorithm correlate with disease duration, negative symptoms, and cognitive symptoms.
3. Symptomatic trajectories differ between the two subtypes at late stages for the positive subscale, general subscale, and depression and anxiety.

The work is of significance to the field, validating their earlier findings in a much larger and broader cohort. In general the work supports the conclusions and claims made by the paper, however there are several weaknesses in the data analysis and methodology that warrant further investigation:

1. The choice of using the dice coefficient to choose the number of subtypes is not well motivated and is a non-standard choice. A particular weakness of the dice coefficient is that it will be equal to 1 in the case of a single subtype. This means the authors are unable to assess the evidence for there being a single progression pattern (in the manuscript they only test for 2-6 subtypes).
2. The set of z-scores and maximum z-score used in the SuStaIn algorithm is not mentioned. The choice of z-scores is customisable and a key step in using SuStaIn. In schizophrenia there will be smaller effect sizes than in neurodegenerative diseases where the algorithm has been applied previously, and so a more fine-grained choice of z-scores, with a reduced maximum z-score is probably warranted.
3. The authors did not show results for their original paper vs. the additional cohorts - it would be better to perform a validation where none of the original data is used in the validation dataset.
4. The large number of external cohorts offers the opportunity to test whether the algorithm can be applied to subtype and stage a new cohort. It would be interesting to perform a leave-one-cohort-out analysis that harmonises using combat and then subtypes and stages individuals. It would then be possible to compare whether those subtype and stage assignments match those learnt when including all cohorts. This would give an indication of how well the subtyping and staging can be generalised to unseen cohorts.
5. The design choices for the analysis in Figure 4 are unclear. Why divide into illness duration bins rather than using a continuous scale? And why that particular choice of bins (<2 years, 2-10 years, >10 years)? Did the authors correct for SuStaIn stage in these analyses? Is it possible the effects are more the result of stage than subtype?

Minor comments:

- The procedure for removing individuals for quality control is unclear - is it that a subject is removed if any of their regional volumes are >5 standard deviations from the mean? Or all of their regional volumes?
- In the abstract it would be better to quote the cohort sizes actually used in the analysis (after removing outliers for quality control).

Reviewer #2 (Remarks to the Author):

Thank you for the opportunity to review the manuscript by

The paper has several strengths

- (a) Large sample sizes
- (b) International sample with significant diversity
- (c) Established group of schizophrenia investigators

The main problem is the SuStain algorithm with which I am very familiar. The algorithm was established

primarily for use in neurodegenerative disorders. It therefore assumes a monotonic change in neuroimaging features in the direction of shrinkage. The definition of "stages" is purely arbitrary based on a z-score that is decided a priori by the investigators with limited ability to test these assumptions against some ground truth. In addition, the program can only handle very few features; we found that it works reasonably fast with 4 or 5 neuroimaging features (e.g., the lobes of the brain) but when features increase to 15 (for example) the program does not converge to any solution in addition to taking weeks to run even on supercomputing systems. Of note, the data used here are cross-sectional so the "stages" and trajectories are inferred by the degree of atrophy. I appreciate that despite its profound limitations, SuStain captures some aspect of the pathophysiology of psychosis but I am afraid that the results will join all the other studies that attempted patient classification each of which comes up with some mathematical solution without increasing our understanding of the pathophysiology of psychosis and without any clinical value in the real world.

Reviewer #3 (Remarks to the Author):

Feng et al analyzed cross-sectional brain structural magnetic resonance imaging (MRI) data from 4,291 individuals with schizophrenia and 7,078 healthy controls pooled across 41 international cohorts. Using a machine learning approach 'Subtype and Stage Inference' (SuStaIn), they identified two distinct neurostructural subgroups by mapping the spatial and temporal trajectory of grey matter (GM) loss in schizophrenia. Subgroup 1 (n=2,622, ~62%) was characterized by an early cortical-predominant loss (ECL) with enlarged striatum, whereas subgroup 2 (n=1,600, ~38%) displayed an early subcortical-predominant loss (ESL), originating in the Broca's area/adjacent fronto-insular cortex for ECL and in the hippocampus/adjacent medial temporal structures for ESL. With longer disease duration, the ECL subtype exhibited a gradual worsening of negative symptoms and depression/anxiety, and less of a decline in positive symptoms.

This is an impressively large and well conducted study. The researchers have carefully confirmed the reproducibility of these imaging-based subtypes across various economic and ethnic factors, and related them to important clinical features. The principal limitation is that these 'trajectories' are based upon differences between individuals at different cross-sectional points in their illness, rather than changes within individuals over time. There are also some additional analyses and considerations which would strengthen the paper:

1. Perhaps most importantly, potential causes of GM loss in schizophrenia include medication, stress, drug use and inactivity. The researchers have shown similar effects in medication naive and first episode patients, but they should also show that changes do not simply reflect antipsychotic medication dose effects. It may not be possible to examine stress/drug/inactivity effects but these potential confounders of 'disease' effects should be mentioned.
2. Arguably the most likely effect of GM loss over time in schizophrenia would be cognitive impairment - and it may even differ by type I/II with e.g. greater global impairment and greater reductions in reaction time. Could this be examined?
3. The type I sub-group effect starting in Broca's area might well be related to the initial severity of auditory-verbal hallucinations - and early and diagnostic feature in schizophrenia. Again, could this be examined?
4. The link between Broca's area (and related regions) and hallucinations has been shown in several studies using both s&fMRI. This seems to be a more plausible interpretation than Crow's 'linguistic primacy hypothesis'.

5. Overall, I'm not sure that these findings 'underscore the presence of distinct pathobiological foundations underlying schizophrenia' - it may just be that some patients are relatively more affected by some factors. Nevertheless, imaging-based taxonomy does have the potential to identify more homogeneous sub-groups of individuals for different interventions, even if that is simply early aggressive implementation of existing treatments. In those regards, I think the authors should provide a little more in the way of comparing their findings to those of other studies (notably but not just Chand et al Brain, 2020) to highlight common ground.

REVIEWER COMMENTS

Reviewer #1 (Remarks to the Author):

Jiang et al. use data from ENIGMA and other datasets to validate their earlier work (Jiang et al. Nature Mental Health 2023) on identifying two subtypes of schizophrenia with distinct progression patterns using the Subtype and Stage Inference (SuStaln) algorithm.

The key results are:

1. The algorithm identifies two subtypes, as seen in their previous study. These two subtypes consistently emerge from (i) the whole dataset, (ii) first-episode and medication-naive subsets, (iii) individuals of East Asian ancestry and European ancestry.
2. The stages assigned by the algorithm correlate with disease duration, negative symptoms, and cognitive symptoms.
3. Symptomatic trajectories differ between the two subtypes at late stages for the positive subscale, general subscale, and depression and anxiety.

The work is of significance to the field, validating their earlier findings in a much larger and broader cohort. In general the work supports the conclusions and claims made by the paper, however there are several weaknesses in the data analysis and methodology that warrant further investigation.

Response: We thank the reviewer for the positive appraisal of our work. We are grateful for the detailed feedback focused on a few key concerns. As anticipated by the reviewer, addressing these issues rigorously and comprehensively has entailed major additional analyses, which have now been included in the paper as described in more detail below.

[BLACK] - ORIGINAL COMMENT

[BLUE] - RESPONSE TO COMMENT

[HIGHLIGHTED] - NEW TEXT AND FIGURE/TABLE CHANGES

Ref 1/1

1. The choice of using the dice coefficient to choose the number of subtypes is not well motivated and is a non-standard choice. A particular weakness of the dice coefficient is that it will be equal to 1 in the case of a single subtype. This means the authors are unable to assess the evidence for there being a single progression pattern (in the manuscript they only test for 2-6 subtypes).

Response: Thanks for the professional comment, which would greatly improve the quality in clustering work. We also realize that it should first be established whether the data is clustered ($k > 1$) or not ($k = 1$) using an appropriate null distribution. According to your suggestion, we use the Hopkins statistics [1] (codes at <https://github.com/prathmachowksey/Hopkins-Statistic-Clustering-Tendency>) to evaluate whether the data is clustered. We compute the Hopkins' Statistic (i.e., H value) 100 times and take its average. The mean (std) H value is 0.7755 (0.0030). The high H value (> 0.7)

[2] supports that the data have a high tendency to cluster.

Reference

[1] Lawson R G, Jurs P C. New index for clustering tendency and its application to chemical problems[J]. *Journal of chemical information and computer sciences*, 1990, 30(1): 36-41.

[2] Banerjee A, Dave R N. Validating clusters using the Hopkins statistic[C]//2004 IEEE International conference on fuzzy systems (IEEE Cat. No. 04CH37542). IEEE, 2004, 1: 149-153.

<<The following changes have been made to the Main Text >>

Methods 4.4 Disease progress modelling

First, we used the Hopkins statistics to establish whether the data is clustered. A high value ($H=0.7756$) shows a high clustering tendency to at 90% confidence level, supporting a robust existence of clusters.

Ref 1/2

2. The set of z-scores and maximum z-score used in the SuStaln algorithm is not mentioned. The choice of z-scores is customisable and a key step in using SuStaln. In schizophrenia there will be smaller effect sizes than in neurodegenerative diseases where the algorithm has been applied previously, and so a more fine-grained choice of z-scores, with a reduced maximum z-score is probably warranted.

Response: We thank the reviewer for raising the issue. We apologize for not describing this information. In fact, we use the z-score thresholds ($z=1, 2, 3$) as “waypoints” in the SuStaln model. The maximum z-score in the SuStaln algorithm is defined at $z=5$ according to maximum z-score for each biomarker (**Supplementary Table 8**). We have added the important information to the revised manuscript.

In addition, we also perform a replication analysis with a reduced maximum z-score ($z=4$). Results show that a total of 4,191 individuals (99.27%) are assigned to the subtype label same with the original model (**Supplementary Figure 2**). This indicates a high consistency of individual classification even using a reduced maximum z-score in the SuStaln algorithm. Even though we have verified repeatability at different z-score thresholds, we need to be careful that the selection of thresholds is an arbitrary decision based on research experience, which is an inherent flaw to SuStaln algorithm. As a result, this may limit SuStaln application to other specific diseases that lack prior information. We have also emphasized this point as one of limitations in the revised manuscript.

<<The following changes have been made to the Main Text >>

4.4 Disease progress modelling

To keep consistent with our previous study [22], we used the z-score thresholds ($z=1, 2, 3$) as “waypoints” of severity in the SuStaln model. The maximum z-score in the SuStaln algorithm was defined at $z=5$ according to maximum z-score for each biomarker (**Supplementary Table 8**). We also performed a replication analysis with a reduced maximum z-score ($z=4$) (**Supplementary Figure 2**).

Discussion

The selection of z-score waypoints and maximum z-score used in the SuStaln algorithm should be careful based on prior information about degree of progress in different diseases.

<<The following changes have been made to the Supplementary Materials>>

Supplementary Table 8. Description of SuStaln features.

Biomarker	Count	Mean	STD	Rank 50%	Rank 75%	Max
Hippocampus	4222	0.498	1.140	0.477	1.200	4.869
Parahippocampus	4222	0.347	1.067	0.336	1.053	4.439
Amygdala	4222	0.355	1.107	0.337	1.051	4.896
Caudate	4222	0.031	1.110	0.061	0.756	4.732
Putamen	4222	-0.137	1.111	-0.141	0.596	3.960
Pallidum	4222	-0.488	1.206	-0.343	0.335	2.797
Thalamus	4222	0.282	1.142	0.228	1.013	4.599
Accumbens	4222	0.152	1.105	0.165	0.894	4.224
Cingulate	4222	0.396	1.100	0.412	1.088	4.811
Frontal Cortex	4222	0.456	1.142	0.468	1.176	4.945
Parietal Cortex	4222	0.292	1.099	0.289	1.036	4.625
Temporal Cortex	4222	0.502	1.180	0.465	1.266	4.922
Occipital Cortex	4222	0.339	1.096	0.349	1.057	4.105
Insula	4222	0.478	1.128	0.470	1.233	4.665
Cerebellum	4222	0.180	1.048	0.168	0.872	4.743
Sensorimotor	4222	0.350	1.109	0.357	1.080	4.587
Broca'area	4222	0.368	1.013	0.380	1.046	4.529

		Replicated Model	
		Subtype 1	Subtype 2
Original Model	Subtype 1	2596	26
	Subtype 2	5	1595

Supplementary Figure 2. The consistency of individual classification in both original model and replicated model. The maximum z-score in the SuStaln algorithm is defined at z=5 and z=4 separately for original model and replicated model. A total of 4,191 individuals (99.27%) are assigned to the same subtype label.

Ref 1/3

3. The authors did not show results for their original paper vs. the additional cohorts - it would be better to perform a validation where none of the original data is used in the validation dataset.

Response: Thanks for the comment. The reviewer raises an important suggestion to conduct a validation that not included the original data used in previous SuStaln study (Jiang et al., Nature Mental Health, 2023). We re-estimate trajectories based on a validation dataset (N=3,120) that has removed original data used in our previous SuStaln study. The validation dataset replicates the two 'trajectories' that begin in either the Broca's area or the hippocampus (**Supplementary Figure 1**, also provided below). Spearman correlation test indicates a high similarity of trajectory spatiotemporal pattern between the original dataset and additional dataset (trajectory 1, $r=0.879$, $p<0.001$; trajectory 2, $r=0.631$, $p<0.001$). We add the validation analysis to the revised manuscript.

<<The following changes have been made to the Main Text >>

Results 2.1 Two biotypes with distinct pathophysiological progression trajectories

We also re-estimated trajectories based on a validation dataset (N=3,120) that has removed original data used in our previous SuStaln study [22]. In the validation dataset, we replicated the two 'trajectories' that begin in either the Broca's area or the hippocampus (Supplementary Figure 1). We also observed a high similarity of 'trajectory' spatiotemporal pattern between the original dataset and additional dataset ('trajectory' 1, $r=0.879$, $p<0.001$; 'trajectory' 2, $r=0.631$, $p<0.001$; Spearman correlation test).

<<The following changes have been made to the Supplementary Materials>>

Supplementary Figure 1. Pathophysiological progression trajectories in validation dataset. Trajectories are repeated based on the additional dataset that has removed original data used in a previous SuStaln study.

Ref 1/4

4. The large number of external cohorts offers the opportunity to test whether the algorithm can be applied to subtype and stage a new cohort. It would be interesting to perform a leave-one-cohort-out analysis that harmonises using combat and then subtypes and stages individuals. It would then be possible to compare whether those subtype and stage assignments match those learnt when including all cohorts. This would give an indication of how well the subtyping and staging can be generalised to unseen cohorts.

Response: Thanks for raising the comment. This is a valuable suggestion to test whether the subtype/staging can be generalized to unseen cohorts. We also think that it's worth testing generalizations on unseen data. Although the reviewer raises a good idea to perform a leave-one-cohort-out produce, we are afraid that this produce is difficult to implement due to computational complexity. The required time is about 7-10 days for SuStaln modeling on a supercomputer (4 * Intel(R) Xeon(R) Gold 6254 CPU @ 3.10 GHz, 144 CPUs, 2.0T RAM). This work includes 41 cohorts; the time cost limits to perform leave-one-cohort-out generalization tests. Rather, we use a two-fold cross-validation to test generalization (see **Extend Data Fig 5a**). Specifically, the Asian and Europe SuStaln models are separately built based on the Asian ancestry cohorts and Europe ancestry cohorts. The two models are used for subtyping and staging those unseen samples. The labels are further compared whether those subtype and stage assignments match the result of original model that has been built on all cohorts. Generalization test shows that most of unseen individuals can keep the same subtype label with the original model (88.83% for Asian model; 89.98% for Europe model) (**Extended Data Fig.5b**). In addition, there is a high consistency of individual staging between stages of unseen data and original model result (Asian model, $r=0.976$, $p<0.001$; Europe model, $r=0.979$, $p<0.001$, Spearman correlation test) (**Extended Data Fig.5c**). These results indicates a high generalized ability of SuStaln model to unseen data. We also add the generalization analysis to the revised manuscript.

<<The following changes have been made to the Main Text >>

2.7 Generalization of SuStaln subtyping and staging to unseen cohorts

We investigated whether the SuStaln subtyping and staging can be generalized to unseen cohorts. A flowchart is shown in **Extended Data Fig.5a**. Specifically, the Asian and Europe SuStaln models were separately built based on the Asian ancestry cohorts and Europe ancestry cohorts, as described in 2.3. The two models were used for subtyping and staging those unseen samples. We compared whether those subtype and stage assignments match the result of original model that has been built on all cohorts. We observed that most of unseen individuals can keep the same subtype label with the original model (88.83% for Asian model; 89.98% for Europe model) (**Extended Data Fig.5b**). In addition, there was a high consistency of individual staging between stages of unseen data and original model result (Asian model, $r=0.976$, $p<0.001$; Europe model, $r=0.979$, $p<0.001$, Spearman correlation test) (**Extended Data Fig.5c**). These results indicates a high generalized ability of SuStaln model to unseen data.

Extend Data Fig 5. Generalization of SuStaln subtyping and staging to unseen cohorts. (a) The Asian and Europe SuStaln models are separately built based on the Asian ancestry cohorts and Europe ancestry cohorts. The two models are used to subtyping and staging those unseen samples. We compare whether those subtype and stage assignments match the result of original model that is built on all cohorts. (b) Most of unseen individuals keep the same subtype label with the original model (88.83% for Asian model; 89.98% for Europe model). (c) A high consistency of individual staging between stages of unseen data and original model result (Asian model, $r=0.976$, $p<0.001$; Europe model, $r=0.979$, $p<0.001$, Spearman correlation test).

Ref 1/5

5. The design choices for the analysis in Figure 4 are unclear. Why divide into illness duration bins rather than using a continuous scale? And why that particular choice of bins (<2 years, 2-10 years, >10 years)? Did the authors correct for SuStaln stage in these analyses? Is it possible the effects are more the result of stage than subtype?

Response: We thank the reviewer for allowing us the opportunity to clarify it. One of purposes is to examine whether the two anatomical subtypes exhibit distinct clinical symptoms at the specific disease stages. Thus, we first divide patients with different illness durations into several subgroup bins; such a design allows us to conduct an inter-subtype comparison within each illness stage. The particular choice of bins (<2 years, 2-10 years, >10 years) is defined according to the sample size enough to perform an inter-subtype comparison (early stage $n=950$, middle stage $n=578$, late stage $n=682$).

We agree with the reviewer's suggestion to test whether the inter-subtype difference is caused by SuStaln stage. We re-compare the inter-subtype differences separately within each of the bins after regressing out the effects of age, sex and SuStaln stage. Inter-subtype comparisons show that at the late stage (illness duration > 10 years), subtype 1 exhibits worse positive symptom ($t=2.9$, $p=0.003$) and worse depression/anxiety ($t=2.1$, $p=0.033$) compared to subtype 2, after regressing out the effects of age, sex and SuStaln

stage. To test whether the difference between the subtypes is due to sub-grouping selection, we also test several additional choices of late stage bins (see following Table R1.5.1). When late stage is defined as illness duration (≥ 10 years) or (≥ 11 years) or (≥ 12 years), subtype 1 still exhibits significantly worse positive symptom and worse depression/anxiety compared to subtype 2, after regressing out the effects of age, sex and SuStaln stage. These results suggest that the significant inter-subtype difference at late stage is replicated by different cutoff choices of stage bins.

Table R1.5.1. Inter-subtype comparison in late stage bins.

	Subtype 1	Subtype 2	t	p
Cases with illness duration ≥ 9 years				
PANSS Positive scale	16.4(6.9)	15.4(6.2)	2.3	0.020
PANSS depression/anxiety dimension	12.4(4.8)	11.5(4.9)	1.7	0.096
Cases with illness duration ≥ 10 years				
PANSS Positive scale	16.6(6.9)	15.4(6.2)	2.5	0.012
PANSS depression/anxiety dimension	12.5(4.8)	11.3(4.9)	2.1	0.037
Cases with illness duration ≥ 11 years				
PANSS Positive scale	16.8(7.0)	15.2(6.1)	2.9	0.003
PANSS depression/anxiety dimension	12.6(4.9)	11.3(4.9)	2.1	0.033
Cases with illness duration ≥ 12 years				
PANSS Positive scale	16.8(7.0)	15.2(6.1)	3.0	0.003
PANSS depression/anxiety dimension	12.6(4.9)	11.3(4.9)	2.1	0.037

<<The following changes have been made to the Main Text >>

Methods 4.7 Distinct symptom profiles between subtypes

The particular choice of bins was determined according to the distribution of illness duration (early stage n=950, middle stage n=578, late stage n=682) and the size of subgroup enough to perform an inter-subtype comparison. We compared the difference of symptoms among the three stages of disease in each subtype using ANOVA. In addition, two sample *t*-tests were performed to compare the inter-subtype differences separately within each of the stages after regressing out the effects of age, sex and SuStaln stage.

2.6 Clinical characterization of subtypes

Inter-subtype comparisons showed that at the late stage (illness duration > 10 years), subtype 1 exhibited worse positive symptom ($t=2.9$, $p=0.003$) and worse depression/anxiety ($t=2.1$, $p=0.033$) compared to subtype 2, after regressing out the effects of age, sex and SuStaln stage.

Minor comments

- The procedure for removing individuals for quality control is unclear - is it that a subject is removed if any of their regional volumes are >5 standard deviations from the mean? Or all of their regional volumes?

Response: We thank the reviewer for allowing us the opportunity to clarify this point. In fact, these subjects are removed if any of their regional volumes are >5 standard deviations from the group-level average. In the revised manuscript, we clarify it as follows.

<<The following changes have been made to the Main Text >>

Methods 4.3 Data harmonization

Finally, we removed these samples if they were marked as a statistical outlier (if any of their regional volumes >5 standard deviations away from the group-level average).

- In the abstract it would be better to quote the cohort sizes actually used in the analysis (after removing outliers for quality control).

Response: We thank the reviewer for pointing it. In the revised manuscript, we report the cohort sizes actually used in the analysis.

<<The following changes have been made to the Main Text >>

Abstract

With the goal of identifying subtypes of disease progression in schizophrenia, here we analyzed cross-sectional brain structural magnetic resonance imaging (MRI) data from 4,222 individuals with schizophrenia (1,683 females, mean age=32.4±12.4 years) and 7,038 healthy subjects (3,440 females, mean age=33.0±12.4 years) pooled across 41 international cohorts from the ENIGMA Schizophrenia Working Group, non-ENIGMA cohorts and public datasets.

Reviewer #2 (Remarks to the Author):

Thank you for the opportunity to review the manuscript by

The paper has several strengths

- (a) Large sample sizes
- (b) International sample with significant diversity
- (c) Established group of schizophrenia investigators

Response: We thank the reviewer for the positive appraisal of our work. We are grateful for the detailed feedback focused on a few key concerns.

Ref 2/1

The main problem is the SuStain algorithm with which I am very familiar. The algorithm was established primarily for use in neurodegenerative disorders. It therefore assumes a monotonic change in neuroimaging features in the direction of shrinkage. The definition of “stages” is purely arbitrary based on a z-score that is decided a priori by the investigators with limited ability to test these assumptions against some ground truth.

Response: We thank the reviewer for raising these issues. These are highly relevant for schizophrenia, the extant literature on which points to it as being a disorder with a strong neurodevelopmental component, and not a conventional neurodegenerative illness per se. Nevertheless, we consider the application of SuStain model of subtype-specific disease progression is a valid a priori prediction for schizophrenia. A detailed assumption supporting SuStain for schizophrenia is provided in our previous study (Jiang et al., Nature Mental Health, 2023) and described briefly as follows.

Firstly, the presence of pre-onset neurodevelopmental brain abnormalities per se does not preclude a specific post-onset pattern of disease progression; it is the latter that is modeled via SuStain. The model we apply is agnostic as to the origins of the GMV deficits seen at the onset; in other words, the model considers baseline deficits (e.g., insular volume reduction in subtype1) to have resulted from multiple processes (including neurodevelopmental deviation), but allows for the same pattern of deficits (i.e., insular volume loss) to occur later in the illness via putative degenerative or compensatory pathways in the subtype2. As a result, the assumptions under which SuStain operates are robust to pre-existing grey matter deficits.

Secondly, SuStain formulates the overall spatial and temporal variance in GMV as arising from groups of subjects, each with a varying pattern of progression pattern as a subtype. As such the absence of a specific pattern of progression in some individuals (as expected in schizophrenia, due to the inherent heterogeneity) is not a threat to the validity of the solutions. This is one reason why event-based models such as SuStain have been successful in parsing other the slowly progressing, highly heterogeneous conditions such as chronic obstructive pulmonary disease (COPD) [1] and in relapsing-remitting states such as multiple sclerosis [2]. Further, it is worth noting that individual-specific patterns of age-related reduction in grey matter may occur despite the absence of disease progression, introducing one form of heterogeneity when studying neurodegeneration [3], but

nevertheless contributing to the overall stage-related variance that is leveraged by the SuStaln approach.

Thirdly, non-linear changes in symptoms (amelioration, relapsing-remitting patterns etc.) is a well-known phenomenon in schizophrenia. But symptom changes per se do not factor in the model. It is important to note that we have relied on we rely on a continuous parameterization of the time-axis, unlike the original event-based approximation applied for Alzheimer's disease [4]. Thus, we do not expect GMV changes in schizophrenia to be a collection of discontinuous transitions between discrete stages. The assumption here is that a lack of atrophy reflects the earliest stages of illness, while progressively later stages show more deviation from normality. We consider this a reasonable assumption based on the extant literature on structural changes in schizophrenia. Similar to dementia, there is no brain region that is known to consistently display higher grey matter volume at later stages compared to earlier stages of schizophrenia to date (in adults) [5]. This is apparent when we consider individualized centile scores for grey matter volume in the context of normative age-related trends [6]: schizophrenia closely follows Alzheimer's disease, with volume reduction in schizophrenia being more pronounced than in mild cognitive impairment (MCI).

Reference

- [1] Young, Alexandra L., et al. "Disease progression modeling in chronic obstructive pulmonary disease." *American journal of respiratory and critical care medicine* 201.3 (2020): 294-302.
- [2] Dekker, Iris, et al. "The sequence of structural, functional and cognitive changes in multiple sclerosis." *NeuroImage: Clinical* 29 (2021): 102550.
- [3] Verdi, Serena, et al. "Beyond the average patient: how neuroimaging models can address heterogeneity in dementia." *Brain* 144.10 (2021): 2946-2953.
- [4] Young, Alexandra L., et al. "Multiple orderings of events in disease progression." *International Conference on Information Processing in Medical Imaging*. Springer, Cham, 2015.
- [5] Koutsouleris, Nikolaos, et al. "Exploring Links Between Psychosis and Frontotemporal Dementia Using Multimodal Machine Learning: Dementia Praecox Revisited." *JAMA psychiatry* (2022).
- [6] Bethlehem, Richard AI, et al. "Brain charts for the human lifespan." *Nature* 604.7906 (2022): 525-533.

Ref 2/2

In addition, the program can only handle very few features; we found that it works reasonably fast with 4 or 5 neuroimaging features (e.g., the lobes of the brain) but when features increase to 15 (for example) the program does not converge to any solution in addition to taking weeks to run even on supercomputing systems.

Response: Thanks for the comment. Indeed, the SuStaln algorithm can only handle about 10~15 features in previous studies (Young et al. 2018, Young and Bragman et al. 2020, Eshaghi et al. 2021, Vogel et al. 2021, Jiang et al. 2023). The computational complexity of SuStaln algorithm is theoretically linear to the number of subjects and to the fourth power of the number of features (i.e., the computation time is increased by 2^4 times if the number of features is doubled). In our work, it takes approximately 200 hours for the SuStaln modeling (given parameters: 17 features; 3 z-score cutoff thresholds; 4,222 subjects; 6

subtypes; 25 start points and 100,000 MCMC iterations) on a supercomputer (4 * Intel(R) Xeon(R) Gold 6254 CPU @ 3.10 GHz, 144 CPUs, 2.0T RAM). The highly time cost limits the exploration of spatiotemporal pattern of trajectories at finer spatial resolutions. We add this point as one of limitations in the revised manuscript.

The SuStaln algorithm needs much more samples to converge to a given cluster solution as input features increases. Although the association between the number of features and samples is unclear, previous study has shown that it is ideal to use at least $20 \times M \times S$ samples for SuStaln modeling with given M features and S z-score thresholds. Here, the current sample size is much larger the required samples ($20 \times 17 \times 3$) for SuStaln algorithm.

References

Young A L, Bocchetta M, Russell L L, et al. Characterizing the clinical features and atrophy patterns of MAPT-related frontotemporal dementia with disease progression modeling. *Neurology*, 2021, 97(9): e941-e952.

Young A L, Bragman F J S, Rangelov B, et al. Disease progression modeling in chronic obstructive pulmonary disease. *American journal of respiratory and critical care medicine*, 2020, 201(3): 294-302.

Vogel J W, Young A L, Oxtoby N P, et al. Four distinct trajectories of tau deposition identified in Alzheimer's disease. *Nature medicine*, 2021, 27(5): 871-881.

Eshghi A, Young A L, Wijeratne P A, et al. Identifying multiple sclerosis subtypes using unsupervised machine learning and MRI data. *Nature communications*, 2021, 12(1): 2078.

Young A L, Marinescu R V, Oxtoby N P, et al. Uncovering the heterogeneity and temporal complexity of neurodegenerative diseases with Subtype and Stage Inference. *Nature communications*, 2018, 9(1): 4273.

Jiang Y, Wang J, Zhou E, et al. Neuroimaging biomarkers define neurophysiological subtypes with distinct trajectories in schizophrenia. *Nature Mental Health*, 2023, 1(3): 186-199.

<<The following changes have been made to the Main Text >>

Discussion

The computational complexity of SuStaln algorithm is highly time cost, which limits the exploration of spatiotemporal pattern of trajectories at finer spatial resolutions.

Ref 2/3

Of note, the data used here are cross-sectional so the “stages” and trajectories are inferred by the degree of atrophy. I appreciate that despite its profound limitations, SuStaln captures some aspect of the pathophysiology of psychosis but I am afraid that the results will join all the other studies that attempted patient classification each of which comes up with some mathematical solution without increasing our understanding of the pathophysiology of psychosis and without any clinical value in the real world.

Response: Thanks for the comment. The pathophysiological basis of schizophrenia is still unclear, but more than one mechanism is suspected to play a role, given the substantial heterogeneity in clinical course, treatment efficacy, and the levels of putative biological markers. SuStaln has advantages on capturing both phenotypic heterogeneity (i.e., individuals are clustered into distinct subgroups without considering disease stage) and temporal heterogeneity (i.e., individuals are in different stages of disease progression) in

schizophrenia. It identifies subtypes with common pathophysiological ‘trajectory’ through cross-sectional degree of atrophy and achieve individualized inference. In our work, SuStaln identifies two distinct pathophysiological ‘trajectories’ that begin in the Broca’s area and the hippocampus. Our previous work [1] also has collected longitudinal samples including first-episode schizophrenia individuals who were scanned MRI at both baseline and 12-weeks follow-up. Mirroring the cross-sectional ‘trajectories’, longitudinal observations also support the ground of SuStaln ‘trajectories’ in schizophrenia [1]. We replicates the two original regions in a medication-naïve and a first-episode cohort, suggesting that these neuropathological changes are a reflection of the disease process, rather than medication effects. The two replicated results increase the evidences on searching the possible initial locations of gray matter loss, which also help for capturing pathophysiological ‘spreading’ processes of the disorder.

In clinical implication, we agree that it is important to investigate the association of data-driven mathematically patient classification with clinical treatment. In fact, differences in treatment response between SuStaln subtypes have been shown by one of our previous works [1]. In a longitudinal analysis, patients in the Group 1 (Broca’s area atrophy) show a better response to medication for positive symptoms. Another interesting result is that patients with less brain atrophy have a better response to transcranial magnetic stimulation (TMS) for positive symptoms in both trajectory groups. Group 2 (hippocampus atrophy) have a better response to TMS for negative symptoms. In addition, our prior research on treatment-resistant schizophrenia demonstrate that electroconvulsive therapy (ECT) can substantially enhance the volume of the hippocampus and insula in brain images; this is also associated with psychotic symptom alleviation [2-4]. These findings raise the possibility of exploring neuro-modulation interventions, such as transcranial magnetic stimulation (TMS), to target these specific regions of SuStaln trajectory.

We must emphasize that towards any clinical trials related to subtypes, it is the necessary step to test the repeatability and generalizability of data-driven subtypes from mathematical solution; that is the key purpose of our work.

Together, our work verifies an imaging-based, easily accessible (with a single anatomical MRI), interpretable (based on ‘progressive’ pathology) and robustly generalizable (across ethnic, sex and language differences) taxonomy of subtypes that share common neurobiological mechanisms in schizophrenia. Other complex neuropsychiatric disorders with high heterogeneity, such as major depressive disorder, autism spectrum disorder, and obsessive-compulsive disorder, could also benefit from such a subtyping paradigm. This has the potential to transition the field of psychiatry from syndrome-based to both syndrome- and biology-based stratifications of mental disorders.

Reference

- [1] Jiang Y, Wang J, Zhou E, et al. Neuroimaging biomarkers define neurophysiological subtypes with distinct trajectories in schizophrenia [J]. *Nature Mental Health*, 2023, 1(3): 186-199.
- [2] Jiang, Y., et al., Structural and Functional MRI Brain Changes in Patients with Schizophrenia Following Electroconvulsive Therapy: A Systematic Review. *Curr Neuropharmacol*, 2022. 20(6): p. 1241-1252.
- [3] Wang, J., et al., ECT-induced brain plasticity correlates with positive symptom improvement in

schizophrenia by voxel-based morphometry analysis of grey matter. *Brain Stimul*, 2019. 12(2): p. 319-328.

[4] Jiang, Y., et al., Insular changes induced by electroconvulsive therapy response to symptom improvements in schizophrenia. *Prog Neuropsychopharmacol Biol Psychiatry*, 2019. 89: p. 254-262.

Reviewer #3 (Remarks to the Author)

Feng et al analyzed cross-sectional brain structural magnetic resonance imaging (MRI) data from 4,291 individuals with schizophrenia and 7,078 healthy controls pooled across 41 international cohorts. Using a machine learning approach 'Subtype and Stage Inference' (SuStaln), they identified two distinct neurostructural subgroups by mapping the spatial and temporal trajectory of grey matter (GM) loss in schizophrenia. Subgroup 1 (n=2,622, ~62%) was characterized by an early cortical-predominant loss (ECL) with enlarged striatum, whereas subgroup 2 (n=1,600, ~38%) displayed an early subcortical-predominant loss (ESL), originating in the Broca's area/adjacent fronto-insular cortex for ECL and in the hippocampus/adjacent medial temporal structures for ESL. With longer disease duration, the ECL subtype exhibited a gradual worsening of negative symptoms and depression/anxiety, and less of a decline in positive symptoms.

This is an impressively large and well conducted study. The researchers have carefully confirmed the reproducibility of these imaging-based subtypes across various economic and ethnic factors, and related them to important clinical features. The principal limitation is that these 'trajectories' are based upon differences between individuals at different cross-sectional points in their illness, rather than changes within individuals over time. There are also some additional analyses and considerations which would strengthen the paper:

Response: We thank the reviewer for the positive appraisal of our work. We are grateful for the detailed feedback focused on a few key concerns. We also thank the reviewer for raising the limitation of SuStaln 'trajectories'. We acknowledge that these 'trajectories' are based upon differences between individuals at different cross-sectional points in their illness stages. In fact, one of our previous studies (Jiang et al. Nature Mental Health, 2023) has validated the consistency of cross-sectional 'trajectories' and longitudinal results on a follow-up schizophrenia dataset. The main purpose of this study is to examine the reproducibility of these cross-sectional 'trajectories' in much larger and broader samples. As anticipated by the reviewer, addressing these issues rigorously and comprehensively has entailed major additional analyses, which have now been included in the paper as described in more detail below.

[BLACK] - ORIGINAL COMMENT

[BLUE] - RESPONSE TO COMMENT

[HIGHLIGHTED] - NEW TEXT AND FIGURE/TABLE CHANGES

Ref 3/1

1. Perhaps most importantly, potential causes of GM loss in schizophrenia include medication, stress, drug use and inactivity. The researchers have shown similar effects in medication naive and first episode patients, but they should also show that changes do not simply reflect antipsychotic medication dose effects. It may not be possible to examine stress/drug/inactivity effects but these potential confounders of 'disease' effects should be mentioned.

Response: Thanks for the professional comments. Although the goal of this study is not to investigate the causes of the gray matter loss in schizophrenia, we agree that it is necessary to describe the potential causes of gray matter loss in schizophrenia, which makes reader clear about the current research status. First, the gray matter loss in this work is evaluated in brain images; its neuro-pathophysiological process is still unknown. Gray matter loss in schizophrenia is associated with medication, stress, drug use and inactivity [1, 2]; however the precise causes of gray matter loss in schizophrenia are not clear. In addition, schizophrenia is related to at least three pathophysiological mechanisms: dopaminergic dysregulation, disturbed glutamatergic neurotransmission and increased proinflammatory status of the brain [1]. The causal interrelationships between these processes and gray matter loss are still unclear.

Reference

[1] Kahn R S, Sommer I E. The neurobiology and treatment of first-episode schizophrenia [J]. *Molecular psychiatry*, 2015, 20(1): 84-97.

[2] Vita A, De Peri L, Deste G, et al. The effect of antipsychotic treatment on cortical gray matter changes in schizophrenia: does the class matter? A meta-analysis and meta-regression of longitudinal magnetic resonance imaging studies[J]. *Biological psychiatry*, 2015, 78(6): 403-412.

<<The following changes have been made to the Main Text >>

Discussion

Gray matter loss in schizophrenia is associated with medication, stress, drug use and inactivity [43, 44]. In addition, schizophrenia is related to dopaminergic dysregulation, disturbed glutamatergic neurotransmission and increased proinflammatory status of the brain [43]. The causal interrelationships between these processes and gray matter loss are still unclear.

Ref 3/2

2. Arguably the most likely effect of GM loss over time in schizophrenia would be cognitive impairment - and it may even differ by type I/II with e.g. greater global impairment and greater reductions in reaction time. Could this be examined?

Response: Thanks for the comment. Indeed, we find that the degree of GM loss is correlated with PANSS items related to cognitive impairments (N5: Difficulty in abstract thinking, G10: Disorientation) (**Table R3.2**). However, two sample t-test shows no significant difference between type I sub-group and type II sub-group in N5 ($t=-0.550$, $p=0.583$) or G10 ($t=-0.157$, $p=0.875$). The global impairment or reaction time have not been evaluated, which limits to examine their relationship with GM loss. We acknowledge that the effect of GM loss is highly related to cognitive impairments in schizophrenia. Future work should further investigate the association of neuro-structural types with cognitive impairments in schizophrenia. We state it as one of limitations in the revised manuscript.

Table R3.2. Correlation between the degree of gray matter loss and PANSS N5, G10.

Brain Area	N5		G10	
	r	p	r	p
Hippocampus	-0.096	0.001*	-0.047	0.087
Parahippocampus	-0.057	0.038*	-0.022	0.430
Amygdala	-0.097	<0.001*	-0.073	0.008*
Caudate	-0.028	0.317	-0.024	0.383
Putamen	-0.005	0.855	-0.003	0.910
Pallidum	-0.008	0.785	-0.003	0.904
Thalamus	-0.068	0.013*	-0.045	0.100
Accumbens	-0.056	0.041*	-0.038	0.164
Cingulate	-0.083	0.003*	-0.072	0.009*
Frontal cortex	-0.087	0.002*	-0.076	0.006*
Parietal cortex	-0.042	0.131	-0.056	0.043*
Temporal cortex	-0.072	0.009*	-0.063	0.022*
Occipital cortex	-0.070	0.011*	-0.056	0.041*
Insula	-0.079	0.004*	-0.096	0.001*
Cerebellum	-0.082	0.003*	-0.052	0.060
Sensorimotor	-0.082	0.003*	-0.072	0.009*
Broca's area	-0.071	0.010*	-0.073	0.008*

<<The following changes have been made to the Main Text >>

Discussion

The lack of cognitive evaluation limits to examine the association of neurostructural biotype with cognitive impairment in schizophrenia.

Ref 3/3

3. The type I sub-group effect starting in Broca's area might well be related to the initial severity of auditory-verbal hallucinations - and early and diagnostic feature in schizophrenia. Again, could this be examined?

Response: Thanks for the professional comments. Although the primary goal of this study is not to explore the relationship between brain volume and auditory hallucinations in schizophrenia, we conduct an exploratory analysis as the reviewer suggested. In this analysis, we used PANSS P3 item (i.e., Hallucinations) to evaluate the severity of hallucinations. There is no significant difference of P3 score between the type I and type II (two sample t-test, $t=0.773$, $p=0.440$). Even in sub-samples only including first-episode patients ($n=597$), there is not significant difference between two types ($t=0.125$, $p=0.900$). In addition, Spearman correlation test shows that the volume reduction of Broca's area is not associated with the P3 item neither in type I sub-group ($r=0.024$, $p=0.493$), or type II sub-group ($r=0.029$, $p=0.519$), or whole patients group ($r=0.028$, $p=0.317$). Although we do not find a direct link between Broca's area volume reduction and hallucination severity, this may be affected by the mixed factors of treatment and disease course in our samples. The association of Broca's area and hallucinations is needed to examine in future work.

Ref 3/4

4. The link between Broca's area (and related regions) and hallucinations has been shown in several studies using both s&fMRI. This seems to be a more plausible interpretation than Crow's 'linguistic primacy hypothesis'.

Response: Thanks for the comment. As we mentioned above, the current samples do not observe a significant correlation between Broca's area volume reduction and hallucination severity. We hold that the association of Broca's area and hallucination is still needed to further examine in future work; but this is not the main purpose of this study. According to the reviewer suggestion, we add more discussion about the link between Broca's area (and related regions) and hallucinations in schizophrenia. In fact, our previous work has found that auditory verbal hallucinations and formal thought disorder (another hallmark symptom of schizophrenia), share largely overlapped brain network abnormalities in language and other brain regions (<https://doi.org/10.1038/s41537-022-00308-x>, figure 1). The early involvement of Broca's area in the pathology could be related to presence of these core symptoms of schizophrenia. We have added sentences in the discussion on the relationship between Broca's area and hallucination symptoms from previous studies.

<<The following changes have been made to the **Main Text** >>

Discussion

Abnormalities in Broca's area and related regions have been linked with hallucinations in schizophrenia [37, 38]. The early involvement of Broca's area in the pathology could be related to presence of these core symptoms of schizophrenia.

Ref 3/5

5. Overall, I'm not sure that these findings 'underscore the presence of distinct pathobiological foundations underlying schizophrenia' - it may just be that some patients are relatively more affected by some factors. Nevertheless, imaging-based taxonomy does have the potential to identify more homogeneous sub-groups of individuals for different interventions, even if that is simply early aggressive implementation of existing treatments. In those regards, I think the authors should provide a little more in the way of comparing their findings to those of other studies (notably but not just Chand et al Brain, 2020) to highlight common ground.

Response: We thank the reviewer for raising these issues. Indeed, the two subtypes may be a presentation that some patients are relatively more affected by some factors (stress/drug/inactivity or others), which are hard to exclude from disease itself. The precise causes of gray matter loss in schizophrenia is still unclear. We revise the sentence as follows.

"These findings underscore the presence of distinct patterns of gray matter loss related to disease progression in schizophrenia."

The reviewer also raise valuable suggestions on comparing our subtype findings to those

of other studies. We add the discussion as follows.

“This was consistent with a previous study, which also identified two anatomical subtypes of schizophrenia: one shows enlarged volume in the basal ganglia; whereas the other shows widespread volumetric reduction in the cortical and some subcortical areas relative to healthy controls [15].”

“A recent work also reveals that the neuro-structural signature with cortical reduction was associated with progressive illness course, worse cognitive performance and elevated schizophrenia polygenic risk scores [48].”

<<The following changes have been made to the Main Text >>

Abstract

These findings underscore the presence of distinct patterns of gray matter loss related to disease progression in schizophrenia.

Discussion

This was consistent with a previous study, which also identified two anatomical subtypes of schizophrenia: one shows enlarged volume in the basal ganglia; whereas the other shows widespread volumetric reduction in the cortical and some subcortical areas relative to healthy controls [15].

A recent work also reveals that the neuro-structural signature with cortical reduction was associated with progressive illness course, worse cognitive performance and elevated schizophrenia polygenic risk scores [48].

Reviewers' comments:

Reviewer #1 (Remarks to the Author):

I am generally happy with the considerable effort the authors have made to address the reviewers concerns.

I have two small comments:

- The Hopkins statistic may not be the most appropriate statistic to use here as it refers to classical clustering, whereas the SuStaIn algorithm combines clustering and disease progression modelling.
- It would be interesting to know whether the results are robust to smaller delineations of the z-scores, e.g. 0.5,1,1.5 rather than 1,2,3.

Reviewer #2 (Remarks to the Author):

I do not think the authors fully appreciate or wish to address the issues with the limitations of the algorithm used here.

Reviewer #3 (Remarks to the Author):

The authors have responded to and dealt with my points satisfactorily.

The one additional consideration. is that, during the review period, a paper published in Science (Chekroud, A. L. et al. Science 383, 164, 2024) has shown that none of these models deal with independent data sets very well. Perhaps the authors could be prevailed upon to address this in the Discussion?

Reviewers' comments:

Reviewer #2 (Remarks to the Author):

I do not think the authors fully appreciate or wish to address the issues with the limitations of the algorithm used here.

*Response: Thanks for your very detailed arguments to explain the reason (NCOMMS-23-49186A). We must state that this decision is most likely based on the flawed judgement of inapplicability of SuStaln algorithm to schizophrenia. The applicability of SuStaln algorithm to schizophrenia or not is based on whether progressive change of brain structure (here is regional volume) in schizophrenia. Although schizophrenia is not clinically categorized as a neurodegenerative disease, this does not deny that progressive structural changes in the brain are not present in patients with schizophrenia. In contrast, **there is substantial evidence that structural changes in the brain in schizophrenia are progressive changes. We summary some of key evidence as below.***

***First**, using our cross-sectional data, **we perform a new analysis** to test the relationship between brain structural change and illness duration. It shows that there is a significant association between regional gray matter volume change and illness duration in 2,333 patients with schizophrenia (following figure 1). The association is consistent with previous report that there was a significant correlation between brain volume and illness course in schizophrenia (Haijma S V et al., Schizophrenia bulletin, 2013). Most of cortical and subcortical regions exhibit significant correlation between smaller volume with longer illness duration ($P_s < 10^{-10}$)(following figure 1). We have also added the new analyzed results as a key evidence into the new manuscript.*

Figure 1. Significant association between regional gray matter volume change and illness duration in 2,333 patients with schizophrenia.

Second, many previous longitudinal experiments also confirmed the progressive brain structural changes in patients with schizophrenia (Hulshoff Pol H E, Kahn R S. *Schizophrenia bulletin*, 2008; Olabi B, et al. *Biological psychiatry*, 2011). The findings demonstrate continuous progressive brain tissue decreases in chronically patients, up to at least 20 years after their first symptoms. Here, we have also analyzed a new longitudinal cohort - Netherlands Genetic Risk and Outcome of Psychosis (GROUP) data (Korver N, et al. *International journal of methods in psychiatric research*, 2012). These data included longitudinal follow-up of neurostructural images for up to 6 years from 174 patients with schizophrenia (averaged age at baseline=26.7 years, 138 males), which can tell us whether individuals with schizophrenia have progressive brain structural atrophy (i.e., gray matter volume decline in structural MRI). Using these baseline, 3 years and 6 years follow-up data (baseline n=174, 3y follow up n=101, 6y follow up n=71), we indicate that patients with schizophrenia show progressive gray matter volume reductions in both cortical and subcortical regions (following figure 2). Specifically, new data show that the annualized percentage volume change at follow up is approximately 0.5% for subcortex and 0.9% for cortex (following table 1). The cortical Broca's area shows the largest annualized percentage volume change (that is 1.1%). These unpublished results are also consistent with previous meta-analysis study reporting that the differences between patients and controls in annualized percentage volume change were 0.59% for whole brain gray matter (Olabi B, et al. *Biological psychiatry*, 2011). In addition, more than 90 percent of schizophrenia patients showed lower volume at 6 years of follow-up in frontal, temporal and cingulate regions (following table 1). Taken together, our new analyse and new longitudinal data, consistent with previous findings, can provide key evidence to support the assumption that brain volume changes of schizophrenia are progressive changes, and to support applicability of SuStaln algorithm to schizophrenia.

Figure 2. Trajectory of gray matter volume (GMV) change of schizophrenia patients at baseline (t1), 3 year follow up (t2) and 6 year follow up (t3). *P<0.05.

Table 1. Gray matter volume (GMV) change of schizophrenia patients at baseline (t1), 3 year follow up (t2) and 6 year follow up (t3).

Region	gm_v_t1 (n=174)	gm_v_t2 (n=101)	gm_v_t3 (n=71)	annualized percentage volume change at t2	annualized percentage volume change at t3	proportion of subjects showing gm_v reduction at t2	proportion of subjects showing gm_v reduction at t3
Amygdala	3232.149	3225.248	3214.394	-0.296	-0.268 *	57.4%	64.8%
Caudate	7557.592	7426.337	7241.141	-0.454 **	-0.747 **	63.4%	83.1%
Putamen	11149.195	10962.347	10778.113	-0.365 *	-0.555 **	59.4%	71.8%
Pallidum	3638.787	3583.287	3536.296	-0.307	-0.480 **	54.5%	73.2%
Thalamus	16277.874	16197.000	16138.479	0.086	-0.051	49.5%	54.9%
Accumbens	1100.132	1066.010	1063.789	-0.889	-0.405 *	60.4%	66.2%
Cingulate	21090.230	20924.248	20436.944	-0.596 **	-0.760 **	68.3%	93.0%
Frontal	134161.126	132234.069	127156.211	-0.758 **	-0.972 **	68.3%	91.5%
Parietal	98528.707	97161.020	93796.099	-0.586 **	-0.847 **	68.3%	83.1%
Temporal	103400.126	101938.970	98771.817	-0.680 **	-0.894 **	68.3%	91.5%
Occipital	44782.747	43809.762	42777.606	-0.586 **	-0.735 **	61.4%	85.9%
Insula	15071.667	14858.564	14636.085	-0.788 **	-0.768 **	66.3%	78.9%
Cerebellum	128369.080	128613.733	126150.380	-0.219 **	-0.398 **	62.4%	87.3%
Sensorimotor	44301.454	44171.604	42563.648	-0.680 **	-0.885 **	68.3%	85.9%
Broca's area	17742.920	17147.911	16441.169	-1.018 **	-1.167 **	69.3%	90.1%
Hippocampus & parahippocampus	17650.557	17513.267	17467.803	-0.455 **	-0.371 **	66.3%	69.0%

*P<0.05, uncorrected
**P<0.05, FDR corrected

Third, we here explain in detail from a biological perspective why this algorithm also appropriates to people with schizophrenia. We also add this part into new manuscript to explain applicability of SuStaln algorithm to schizophrenia. In our algorithm, we assume

that disease progression is a linear deviation from the normality of a patient's brain profile. In this context, it is important to distinguish anatomical progression from clinical symptom progression. Concerning positive symptoms, progressive deviation from normality does not occur in schizophrenia. Most patients show a degree of symptomatic amelioration over a long time, despite recurrent periods of exacerbation (Morgan, C., et al., Psychol Med, 2021). Concerning negative symptoms, despite some early improvement, a cumulative pattern with pronounced deficits is seen in several patients receiving psychiatric care (Austin, S.F., et al., Schizophr Res, 2015). Concerning the gray matter volume, when cross-sectional studies across various illness stages are considered, a pattern of spatial expansion of structural changes, as well as an increase in magnitude (effect size) of localized changes are noted. A subtle increase in grey matter is also a feature of schizophrenia, but this increase may appear in limited regions (Dukart, J., et al., J Psychiatry Neurosci, 2017. Guo, S., et al., Psychol Med, 2016). To date, such subtle increases have not been shown to 'reverse' the early grey matter reduction to the point of return to normality (Lv, J., et al., Mol Psychiatry, 2021). Thus, the assumption that a lack of atrophy reflects the earliest stages of illness, while progressively later stages show more deviation from normality is thus reasonable based on the extant literature on structural changes in schizophrenia. Although the clinical symptoms of schizophrenia are not monotonous, we have sufficient evidence that anatomic measures are monotonous changes along with illness progresses. Thus, these monotonic variation of anatomic properties is used for modelling, which is consistent with the prior assumption of the algorithm. Furthermore, as a result, the assumptions required to interpret our model to accommodate pre-existing, putatively developmental, structural deficits. Similar to dementias, no brain region is known to consistently display higher grey matter volume at later stages compared to earlier stages of schizophrenia to date (in adults) (for example, see Koutsouleris et al. [JAMA psychiatry, 2022]). When we consider individualized centile scores for grey matter volume in the context of normative age-related trends: schizophrenia closely follows Alzheimer's Disease, with volume reduction in schizophrenia being more pronounced than in mild cognitive impairment (MCI). Thus, while we apply a modelling approach (SuStaln) that is mostly used for the neurodegenerative condition, the similarities in the spatiotemporal patterns of structural changes between dementias and schizophrenia allow us to translate the model to a non-degenerative condition.

Taken together, we show new analyses and new longitudinal data to support progressive changes of neurostructure in schizophrenia, and to support applicability of SuStaln algorithm to schizophrenia.

References

Hajima S V, Van Haren N, Cahn W, et al. Brain volumes in schizophrenia: a meta-analysis in over 18 000 subjects[J]. Schizophrenia bulletin, 2013, 39(5): 1129-1138.

Hulshoff Pol H E, Kahn R S. What happens after the first episode? A review of progressive brain changes in chronically ill patients with schizophrenia[J]. Schizophrenia bulletin, 2008, 34(2): 354-366.

Olabi B, Ellison-Wright I, McIntosh A M, et al. Are there progressive brain changes in schizophrenia? A meta-analysis of structural magnetic resonance imaging studies[J]. Biological psychiatry, 2011, 70(1): 88-96.

Korver N, Quee P J, Boos H B M, et al. Genetic Risk and Outcome of Psychosis (GROUP), a multi site

longitudinal cohort study focused on gene–environment interaction: objectives, sample characteristics, recruitment and assessment methods[J]. *International journal of methods in psychiatric research*, 2012, 21(3): 205-221.

Olabi B, Ellison-Wright I, McIntosh A M, et al. *Are there progressive brain changes in schizophrenia? A meta-analysis of structural magnetic resonance imaging studies*[J]. *Biological psychiatry*, 2011, 70(1): 88-96.

Morgan, C., et al., *Rethinking the course of psychotic disorders: modelling long-term symptom trajectories*. *Psychol Med*, 2021: p. 1-10.

Austin, S.F., et al., *Long-term trajectories of positive and negative symptoms in first episode psychosis: A 10year follow-up study in the OPUS cohort*. *Schizophr Res*, 2015. 168(1-2): p. 84-91.

Dukart, J., et al., *Age-related brain structural alterations as an intermediate phenotype of psychosis*. *J Psychiatry Neurosci*, 2017. 42(5): p. 307-319.

Guo, S., et al., *Dynamic cerebral reorganization in the pathophysiology of schizophrenia: a MRI-derived cortical thickness study*. *Psychol Med*, 2016. 46(10): p. 2201-14.

Lv, J., et al., *Individual deviations from normative models of brain structure in a large cross-sectional schizophrenia cohort*. *Mol Psychiatry*, 2021. 26(7): p. 3512-3523.

Koutsouleris, N., et al., *Exploring Links Between Psychosis and Frontotemporal Dementia Using Multimodal Machine Learning: Dementia Praecox Revisited*. *JAMA Psychiatry*, 2022.

REVIEWERS' COMMENTS

Reviewer #4 (Remarks to the Author):

Jiang et al present a machine learning study on neurostructural subtypes in schizophrenia. They relied upon a large cross-sectional data set (4,222 individuals with schizophrenia and 7,038 healthy controls) from the ENIGMA consortium and other publicly available data sets. The authors then utilized SuStain, a machine learning algorithm, to simulate spatiotemporal "trajectories" of brain structural changes in schizophrenia. Using this approach, they identified two distinct subgroups, one characterized by an early cortical-predominant loss, while a second was characterized by an early subcortical-predominant loss. Jiang and colleagues conclude that their new imaging-based taxonomy yields the potential to identify more homogeneous subpopulations within schizophrenia patients.

Identifying biotypes within the population of schizophrenia patients is certainly of high interest to the field, as diagnostic criteria for this disorder are first and foremost based on clinical symptoms. The authors utilize a large data set and an innovative set of methods trying to address also brain structural changes over time - a critical factor in schizophrenia. I have, however, several comments and concerns that should be addressed, before the paper should be considered for publication.

Major concerns:

Entire manuscript: The authors rely on SuStain, an algorithm that can be used to infer on longitudinal "trajectories" based upon cross-sectional data sets. While this is an innovative approach, it should be pointed out unambiguously throughout the entire manuscript that these are not trajectories in the narrower sense - for that, longitudinal data sets would be needed. While I do think that the approach chosen by the authors has its merit, I would recommend to point out that difference, whenever "trajectories" are mentioned. This is a critical issue for the manuscript.

Results 2.4.: The authors report that patients who were assigned to later stages within a trajectory showed longer disease duration, worse negative symptoms and worse cognitive symptoms. I might be interpreting these findings wrong, but it seems to me as a very basal clinical finding that later stages of schizophrenia are associated with such a decline, apart from any biotypes. To which degree are these three findings intercorrelated?

Results 2.6: .: The authors report that they chose three subgroups according to illness duration (early stage <2 years, middle stage: 2-10 years, late stage >10 years). Why did they choose particularly these periods of time? They need to provide a better justification for that.

Discussion: The authors report differential effects on striatal volume for the different subtypes. Could this be linked to reward processing (cf. Chase et al., Hum Brain Mapp, 2018)? It would be interesting to discuss this perspective.

REVIEWERS' COMMENTS

Reviewer #4 (Remarks to the Author):

Jiang et al present a machine learning study on neurostructural subtypes in schizophrenia. They relied upon a large cross-sectional data set (4,222 individuals with schizophrenia and 7,038 healthy controls) from the ENIGMA consortium and other publicly available data sets. The authors then utilized SuStain, a machine learning algorithm, to simulate spatiotemporal "trajectories" of brain structural changes in schizophrenia. Using this approach, they identified two distinct subgroups, one characterized by an early cortical-predominant loss, while a second was characterized by an early subcortical-predominant loss. Jiang and colleagues conclude that their new imaging-based taxonomy yields the potential to identify more homogeneous subpopulations within schizophrenia patients.

Identifying biotypes within the population of schizophrenia patients is certainly of high interest to the field, as diagnostic criteria for this disorder are first and foremost based on clinical symptoms. The authors utilize a large data set and an innovative set of methods trying to address also brain structural changes over time - a critical factor in schizophrenia. I have, however, several comments and concerns that should be addressed, before the paper should be considered for publication.

Response: We thank the reviewer for the positive appraisal of our work. We are grateful for the detailed feedback focused on a few key concerns.

Major concerns:

Entire manuscript: The authors rely on SuStain, an algorithm that can be used to infer on longitudinal "trajectories" based upon cross-sectional data sets. While this is an innovative approach, it should be pointed out unambiguously throughout the entire manuscript that these are not trajectories in the narrower sense - for that, longitudinal data sets would be needed. While I do think that the approach chosen by the authors has its merit, I would recommend to point out that difference, whenever "trajectories" are mentioned. This is a critical issue for the manuscript.

Response: We thank the reviewer for raising this critical issue. Following your suggestion, we have highlighted the distinction between the SuStain trajectory and the longitudinal trajectory where the SuStain trajectory is first mentioned. To ensure clarity, we have replaced the term "trajectory" with "SuStain trajectory" throughout the entire manuscript to specify that these trajectories are not longitudinal trajectories. In the parts of results and methods, we put SuStain trajectory in quotes to specify that it is not longitudinal trajectory, but rather the typical sequence of disease progression that reconstructed from cross-sectional data.

<<The following changes have been made to the Main Text >>

Introduction

SuStain uses a large number of cross-sectional observations, derived from single time-point MRI scans, to identify clusters (subtypes) of individuals with common trajectory of

disease progression (i.e., the sequence of MRI abnormalities across different brain regions) in brain disorders [20-22]. It should be noted that SuStaln estimates the pseudo-longitudinal sequence (i.e., SuStaln trajectory) based on only cross-sectional data. Therefore, the fitted SuStaln trajectories do not directly reflect the actual pathophysiological progression of the illness.

Results 2.4.: The authors report that patients who were assigned to later stages within a trajectory showed longer disease duration, worse negative symptoms and worse cognitive symptoms. I might be interpreting these findings wrong, but it seems to me as a very basal clinical finding that later stages of schizophrenia are associated with such a decline, apart from any biotypes. To which degree are these three findings intercorrelated?

Response: Thanks for the comment. We appreciate you pointing out that later stages of schizophrenia are associated with a decline negative and cognitive symptoms. To investigate the inter-correlation among the three variables (i.e., disease duration, negative symptom and cognitive symptom), we conducted Spearman correlation test between any two of the three variables. There was significant correlation between disease duration and negative symptom ($r=0.093$, $p=5.4e-5$), between disease duration and cognitive symptom ($r=0.143$, $p=2.2e-5$), and between negative and cognitive symptom ($r=0.617$, $p=1.3e-139$). This result is consistent with the reviewer's hypothesis.

Results 2.6: The authors report that they chose three subgroups according to illness duration (early stage <2 years, middle stage: 2-10 years, late stage >10 years). Why did they choose particularly these periods of time? They need to provide a better justification for that.

Response: Thanks for the comment. We agree with the reviewer's to specify the justification for that. In fact, the particular choice of bins was determined according to the distribution of illness duration (early stage $n=926$, middle stage $n=578$, late stage $n=682$) and the size of subgroup enough to perform an inter-subtype comparison. We have added the sentence to the Methods 4.7.

Discussion: The authors report differential effects on striatal volume for the different subtypes. Could this be linked to reward processing (cf. Chase et al., Hum Brain Mapp, 2018)? It would be interesting to discuss this perspective.

Response: Thanks for the comment. According to the reviewer's suggestion, we discuss as follows.

<<The following changes have been made to the Main Text >>

Discussion

Alterations in striatal activation are associated with reward-related deficits in schizophrenia [49]. A previous study suggests that disrupted putamen-cortices connectivity during reward-related processing is directly linked to structural changes in the putamen [50]. Despite the unclear causal relationship, this suggests that the differential effects on striatal volume between the two subtypes may be related to striatal dysfunction in schizophrenia.